



# Improved processing and calibration of the interferometric mode of the CryoSat radar altimeter allows height measurements of supraglacial lakes in west Greenland.

Laurence Gray[1], David Burgess[2], Luke Copland[1], Thorben Dunse[3], Kirsty Langley[4], Geir Moholdt[5].

[1]Department of Geography, Environment and Geomatics, University of Ottawa, Ottawa, K1N 6N5, Canada
[2]GSC NRCan, Ottawa, K1A 0E8, Canada
[3]Department of Geosciences, University of Oslo, 0316 Oslo, Norway
[4]Asiaq, Greenland Survey, 3900 Nuuk, Greenland
[5]Norwegian Polar Institute, NO-9296 Tromso, Norway

*Correspondence to*: Laurence Gray (laurence.gray@uottawa.ca)

**Abstract.** We compare geocoded heights derived from the interferometric mode (SARIn) of CryoSat to surface heights from calibration-validation sites on Devon Ice Cap and West Greenland. Comparisons are included for both the heights derived from the first return (the 'point-of-closest-approach' or POCA) as well as heights derived from delayed waveform returns

('swath' processing). While swath processed heights are normally less accurate than edited POCA heights, of order 1 - 5 m instead of order 1- 2 m, the increased coverage possible with swath data complements the POCA data and provides useful information for both system calibration and improving digital elevation models (DEMs). We show that the pre-launch interferometric baseline coupled with an additional roll correction (~0.0075°), or equivalent phase correction (~0.0435 radians), provides an improved calibration of the interferometric SARIn mode.

We extend the potential use of SARIn data by showing the influence of surface conditions, especially melt, on the return waveforms, and that it is possible to detect and measure the height of summer supraglacial lakes in West Greenland. A supraglacial lake can provide a strong radar target in the waveform, stronger than the initial POCA return, if viewed at near normal incidence. This provides an ideal situation for swath processing and we demonstrate height accuracies of ~ 0.5 m for two lake sites, one in the accumulation zone and one in the ablation zone, which were measured every year from 2010 or

2011 to 2016. Each year the lake in the ablation zone was viewed in June by ascending passes and then 5.5 days later by descending passes which allows an approximate estimate of the filling rate. The results suggest that CryoSat waveform data and measurements of supraglacial lake height change could complement the use of optical satellite and be helpful as proxy indicators for surface melt around Greenland.





## 1 Introduction

Temporal change in ice sheet surface elevation derived from satellite altimeters has been used in mass balance estimates and the associated contribution to sea level rise, (e.g. Davis and Ferguson, 2004; Rémy and Parouty, 2009; Shepherd et al., 2012; Hurkmanns et al., 2014; Zwally et al., 2015). Satellite radar altimeters have traditionally operated at Ku band (~ 13 GHz) and

used parabolic transmit/receive dish antennas with a diameter of ~ 1 m, so that the main beam illuminates an area beneath the satellite with a diameter of ~15 km and area of ~180 km$^2$. With a typical bandwidth of ~ 300 MHz the range resolution is ~ 50 cm and, as delay time increases beyond the point at which the first surface returns are received, an increasing area contributes to the received signal. These returns are termed 'pulse limited', with the initial signal originating from the area within the main beam closest to the satellite, often referred to as the 'point-of-closest-approach' (POCA). With these

parameters, the diameter of the initially sampled POCA area over the ocean is ~ 1.2 - 1.5 km, but this isn't necessarily the case over glacial ice. The initial area contributing to the leading edge of the waveform (the delay time variation in received power) over an ice cap depends on the ice cap topography. All we know is that it must originate from somewhere within the area illuminated by the main antenna beam and that part of the POCA surface area must be orthogonal to the incident wave. Considering the large variability in ice cap topography and surface conditions, it is not unexpected that the waveforms from

glacial ice will vary significantly in shape and power. The fact that the geographic position of the POCA is, a priori, unknown is one of the major problems in traditional radar altimetry and methods to get around this limitation have been studied extensively (Brenner et al., 1983, Bamber, 1994, Brenner et al., 2007, Hurkmanns et al., 2012).

The European Space Agency (ESA) launched CryoSat as the first in their 'Earth Explorer' series of satellites, which are designed to explore and demonstrate new techniques and methods in Earth observation. As such, CryoSat was designed to

include a new mode of operation to address some of the limitations of traditional radar altimetry when used over sea ice and ice caps. The new approach borrows heavily from coherent imaging radar technology and uses bursts of pulses in which the frequency of the pulses within each burst is high enough that coherent Doppler processing can be used to focus the energy in the along-track direction and ultimately create a footprint for which the along-track position is known, but the footprint centre can still be displaced from the sub-satellite track dependent on the cross-track slope. The along-track processing

approach is referred to as 'Delay-Doppler' and was pioneered by Raney (1998). The suggestion that cross-track interferometry could solve the cross-track footprint position problem in radar altimetry is due to Jensen (1999). For glacial terrain the new 'SARIn' mode of operation provides a relatively small geocoded footprint which allows, for the first time, a systematic comparison of satellite radar altimeter elevations with surface heights from surface and airborne campaigns.

The first CryoSat satellite equipped with the Synthetic Aperture Interferometric Radar Altimeter (SIRAL) was launched in

2005 but failed to enter orbit. A replacement satellite was launched in 2010 and, as of October 2016, is still operating satisfactorily, three years beyond its design life. CryoSat operates in three modes: a conventional low resolution mode (LRM) which is used over oceans and the interior of Antarctica and Greenland, a synthetic aperture mode (SAR) for use over sea ice, and the interferometric SARIn mode over all the other glacial ice areas on Earth. A comprehensive description



of CryoSat is given by Wingham et al. (2006). Here we are concerned primarily with SARIn mode calibration and with demonstrating some unique capabilities of this new mode of satellite radar altimetry. These depend primarily on the ability to geocode the position of the relatively small footprint.

After the initial commissioning phase of the satellite in spring and summer 2010, intermediate and final products were

available from ESA. For glacial ice the ESA level 2 (L2) product contains the position and height of the geocoded POCA positions. An additional L2i product is available which contains the same geocoded height solution as the L2 product, but also the waveform which can be used to help eliminate poor data and solutions. An intermediate product (L1b) has also been made available which includes the waveform power, phase, coherence, satellite position and velocity, etc., and all the corrections and timing information necessary to calculate the position and height of the POCA footprint. This has been

useful to those users wishing to study processing techniques; for example, by having access to the intermediate L1b product it has been possible to demonstrate that the returns which are time delayed beyond the initial POCA position can be used in areas with suitable cross-track slopes to create 'swath processed' elevations (Gray et al., 2013). Indeed, the L1b products have been used in several studies of change in Antarctica and Greenland (e.g. Helm et al., 2014, Nilsson et al., 2016, Smith et al. 2016), smaller Arctic ice caps (Gray et al. 2015), and lake height (Kleinherenbrink et al. 2014). In these studies, the

authors claim improvements in the results over the standard level 2 product, due to the specialized processing.

Three versions of the various CryoSat products have been distributed by ESA since commissioning; these are the so-called baseline A, B and C products. Details of the improvements can be found through the ESA Earth Online web site devoted to the CryoSat mission (https://earth.esa.int/web/guest/missions/esa-operational-eo-missions/cryosat). Here we have used only the latest baseline C products, particularly because the waveforms in these products span a range window distance of ~ 240 m,

twice the distance available in the baseline B products. Some comparisons are also made between results derived from the baseline C L1b files and those provided in the L2 products.

In this study we use CryoSat and surface height data from two well-studied sites in the Canadian Arctic and Greenland to improve the calibration of the SARIn mode. Further, we show that the waveforms do change significantly with surface melt and that it is possible to detect the formation of supraglacial lakes. By using a modified swath processing scheme, we also

show that it is possible to measure lake height and height change.

## 2 Methods

Our methods in working with the L1b files, were described in Gray et al. (2013) and Gray et al. (2015). The current Matlab processing provides both POCA and swath mode results, and here we note any changes since the earlier work. The method to generate POCA heights are comparable to those described in Helm et al. (2014), Nilsson et al. (2016) and Smith et al

(2016), and were motivated by similar concerns, particularly the performance of the L2 'retracker': this is the algorithm designed to find the position of the POCA return in each waveform.





The Delay-Doppler processing (Raney, 1998) for the SARIn mode of CryoSat is described in Wingham et al. (2006) and Kleinherenbrink et al. (2014). In this method 64 pulses are used in each transmitted burst and fast Fourier transform processing is used to create 64 unfocussed beams so that, with appropriate superposition of results from a sequence of bursts, multiple 'looks' can be averaged for each ground footprint. In practice there are less than 64 looks contributing to each waveform in the L1b file, normally ~ 57. In the along-track direction the footprints are separated by ~ 280 – 300 m and the resolution is ~ 380 m (Bouzinac, 2015). In the cross-track direction the footprint size is dictated by the cross-track slopes and by any smoothing of the waveform in the processing. It is important to note that the position of the POCA footprint derived from each waveform will be in the plane including the satellite position, and the lines defined by the cross-track and nadir directions. The POCA area will be centred on the closest point in the intersection of this plane with the terrain surface so that when ascending and descending orbits cross the two POCA footprints will not be the same when there is a cross-track slope. The L1b files contain two echo scaling parameters for each waveform which allow a calibration of the waveform power to watts, and these have been used in this work to derive the logarithmic (dB) values used in the results.

## 2.1 Selecting the POCA position from the SARIn waveform

If the altimeter response from terrain was 'predictable' it would be beneficial to use the complete waveform in the estimation of the position in delay time of the surface, and this is the basis of the ESA L2 processing. However, our experience with the L1b SARIn waveforms over glacial ice shows that the shape and magnitude of the waveform can vary significantly, even in one area at one time (see examples in section 4). The average return power as a function of delay time from the first surface sample will vary with the illuminated surface area, the reflectivity of the surface and any near surface layering on the ice cap. The cross-track slope and fixed sampling in delay time (3.125 ns) defines the basic cross-track footprint size so that the waveform shape beyond the POCA depends primarily on the variation in topography in the cross-track direction. This is essentially independent of the position of the POCA, hence our decision to estimate the POCA position based on the first significant leading edge in the waveform. Our approach (Gray et al. 2015) uses the point of inflexion (maximum slope) on the first significant waveform increase, and is similar to that adopted by Nilsson et al. (2016) and Smith et al. (2016). Helm et al. (2014) used a threshold level of the first significant leading edge for their work in Greenland and Antarctica, following the work of Davis (1997) who advocated a threshold retracker to minimize the dependency on varying microwave penetration into, and backscattering from, various snow-firn-ice layers. The importance of the cross-track footprint size in dictating the shape of the waveform has been demonstrated by the success of the straightforward waveform simulation based primarily on topography shown in Gray et al. (2013).

Although the L1b waveforms already represent averaged values, some additional smoothing has been done on the complex waveform data. The low-pass filter uses a 3 dB width of ~ 4 samples and is designed to avoid introducing any bias in the waveform phase. Averaging of SARIn waveform data is performed only in the range direction with a relatively small impact on the cross-track footprint size (Gray et al. 2015), and none on the along-track resolution. It is not appropriate to average





any of the L1b waveform data in the azimuth direction because there can be jumps in the delay time to the first waveform sample. Geocoding SARIn data depends on the L1b waveform phase to provide the cross-track look angle. The processing steps to generate geocoded heights are described in Gray et al. (2015) using the results of the calibration described in section 3 below. Solutions are derived for the phase at the estimated POCA position in the waveform, and for this phase $+2\pi$ and -

$2\pi$. Comparison with the height of the reference DEM is used to select the most likely of the three solutions.

## 2.2 Swath mode processing

The techniques used to process the returns delayed beyond the POCA position are essentially as described in Gray et al. (2013). In that work the bias errors associated with the uncertainty in the baseline roll angle (Galin et al. 2013) were reduced by comparing the derived E–W slope on the western flank of Devon Ice Cap with the reference data slope, and changing the

baseline roll angle to minimize this error. This step has not been undertaken here as it presumes a good quality reference DEM which is not necessarily available.

Waveform smoothing can lead to a situation in which results may be oversampled in the cross-track direction. The swath processed results from any one waveform will form a straight line in the cross-track direction and the final samples in cross-track are generated by binning and averaging the results in segments of the cross-track line. The separation between cross-

track samples is nominally ~100 m. Criteria for minimum values of the filtered coherence and returned power are set, and are usually ~0.84 and -150 dB, respectively, for baseline C data. The phase unwrapping and ambiguity checking method is similar to that described by Smith et al. (2016).

The swath processing of the summer CryoSat data for supraglacial lake height (section 4.2) omitted the cross-track binning stage and produced an elevation for each sample in the waveform. Only heights derived from waveform samples with phase

values equivalent to small look angles (< ~ 0.2°), high power (> ~ -140 dB), and high coherence (> 0.95) were used. These minimum values virtually guarantee that there will be a small contribution from the range ambiguous zone, and that phase unwrapping or ambiguity checking is unnecessary. The resulting geographic positions were compared to the best available visible imagery, usually Landsat 8 images, and north, south, east and west boundaries around the lake feature were set. The resulting height estimate was then obtained by averaging all estimates within the lake boundary.

## 2.3 Measuring the height difference between the reference surface and CryoSat heights.

We used two methods to compare the derived CryoSat heights with the surface reference data. For Devon Ice Cap the reference data included inter-calibrated skidoo-based differential GPS transects, and airborne scanning laser altimeter data from both the NASA Airborne Terrain Mapper (ATM; Krabill et al., 2002, Krabill, 2014) and the TUD ALS (https://earth.esa.int/documents/10174/134665/ESA-CryoVEx-ASIRAS-2014-report) systems. For the Greenland site, we

have relied on the ATM data collected on NASA IceBridge flights. The first method stepped through all the CryoSat results and searched for reference heights within 400 m of the centre of the CryoSat footprint. The height differences between the



CryoSat and reference heights were corrected for the slope between the centres of the two footprints using interpolation with the reference DEM. If there were many reference values, as can be the case for the west Greenland site, then a second simpler method was used: a search was made for reference points within 50 m and the height differences were tabulated and averaged without the slope correction stage. Virtually all the reference height data for both sites were obtained under cold

conditions in April or early May and we assumed that any accumulation or change in the backscatter conditions between January and mid-May would lead to a relatively small change in the CryoSat height. This provided the rationale for comparing all the CryoSat results from the January to May passes with the April or May reference height data.

**2.4 Estimating height errors in the CryoSat data**

Ku band radar waves can penetrate the surface and the CryoSat-to-surface height bias will vary depending on the conditions

of the surface and near surface (Gray et al., 2015; Nilsson et al., 2015). Consequently, we use the standard deviation of the height differences about the mean height difference as the primary measure of the quality of the CryoSat measurements. The relatively small error in the ATM or ALS laser surface heights (~ 20 cm, Krabill et al., 2002) is ignored and any impact due to the difference in the footprint size is not considered.

When estimating the height errors for the supraglacial lakes it is not appropriate to quote the standard error (standard

deviation divided by the square root of the number of samples averaged), because the samples will not be independent and there is the possibility of small bias error in the result. The errors were therefore estimated on a case-by-case basis by looking at any cross-track slope across a lake feature, using the standard deviation itself, and checking independent estimates from ascending and descending passes over the same feature. The standard deviation about the mean was typically ~0.5 m, and the mean difference between the ascending and descending passes over the same accumulation zone lake feature in

August was ~ 0.25 m. Table 1 includes the error estimates from two lakes and shows that relatively good accuracy can be achieved for these strong targets, better than the potential error for individual POCA estimates.

**3 Results: SARIn mode calibration**

The key parameters for SARIn mode geocoding are the range to the surface, and the satellite look angle between the normal to the WGS84 ellipsoid and the footprint centre in the cross-track plane. The former involves consideration of timing and the

retracker algorithm for the POCA results, but it is the latter which requires careful calibration for both POCA and swath mode results.

The look angle, $\alpha$, is related to two other angles through:

$$\alpha = \beta - \delta \tag{1}$$



Where $\beta$ is the interferometric angle defined below, and $\delta$ is the roll angle of the interferometric baseline, all defined in the cross-track plane containing the line normal to the WGS84 ellipsoid. The angle $\beta$ is related to the interferometric phase through:

$$\beta = -\mathrm{asin}(ph/kB) \tag{2}$$

5 Where $ph$ is the phase provided in the L1b file, $k$ is the wavenumber, and $B$ is the length of the interferometric baseline. The sense of the look and interferometric angle is as follows: For zero roll an observer siting on the CryoSat satellite facing in the direction of motion with their feet pointing towards the Earth will 'see' a footprint to the right of the sub-satellite track when the look angle $\alpha$ is positive. The roll angle $\delta$ is also provided in L1b files. For the same observer configuration, a positive roll angle corresponds to the left antenna being higher than the right hand one.

10 Any bias in the look angle, $\Delta\alpha$, can then be related to biases in the baseline; $\Delta B$, phase; $\Delta ph$, and roll angle; $\Delta\delta$, through:

$$(\alpha + \Delta\alpha) = -asin\left(\frac{(ph + \Delta ph)}{k(B + \Delta B)}\right) - (\delta + \Delta\delta) \tag{3}$$

Using the approximations that $\sin(x) = x$ for small $x$ and $B \gg \Delta B$, leads to an expression for the bias in roll angle as:

$$\Delta\alpha = \frac{\Delta ph}{kB} - \frac{ph}{kB}\left(\frac{\Delta B}{B}\right) - \Delta\delta \tag{4}$$

The CryoSat satellite and processing chain contains careful controls which should minimize any extraneous inter-channel 15 phase shift $\Delta ph$ on the satellite (Bouzinac, 2015). Even if a residual phase bias exists, due perhaps to an uncompensated path length difference between the two receivers, it can be expressed in the same form as the roll angle correction $\Delta\delta$ and the two can be considered together. The second term in equation (4) reflects the possibility of a bias between the actual and pre-launch measurement of the interferometric baseline; the distance between the two antenna phase centres. This was part of the post-launch SARIn mode calibration carried out by Galin et. al. (2013). This work used satellite roll manoeuvres over mid 20 latitude ocean tracks to present evidence that the interferometric angle should be scaled by a factor of $0.973 \pm 0.002$, which is equivalent to scaling the baseline by a factor of 1.0277. The third term in Eq. (4), the uncertainty in the baseline roll angle $\Delta\delta$, is potentially the most important because, as documented by Galin et. al. (2012), the baseline roll angle is derived from one of three star trackers mounted on a support bench on the satellite which apparently bends. Consequently, there is an unknown bias in the reported value of the baseline roll angle which can vary pass-to-pass, presumably as the satellite 25 experiences a varying history of solar illumination. Here we use SARIn data over well-documented glacial ice to investigate any residual bias in the roll angle provided in the L1b files, and to study the influence of changing the baseline length in processing L1b files.

## 3.1 Calibration test sites

We use data from two sites, the western flank of Devon Ice Cap (Fig 1), and an area in western Greenland including the 30 Jakobshavn Glacier (Fig. 2), as both have excellent reference surface height data. Our calibration approach depends on the



presence of a predominantly east-west slope which is why the test area in Fig. 1 is limited in the north-south direction. By using terrain with an east-west slope we obviate the necessity for roll-tilting the satellite. Figure 3 illustrates the difference in the nature of the slopes for the two test sites. The significant increase in slope variation in the west Greenland site represents a more challenging situation for satellite radar altimetry than the more modest slope variation on the western flank of Devon

Ice Cap, and this is why we have concentrated on comparing the results from these two test sites.

## 3.2 Calibration based on data from Devon Ice Cap

The western portion of Devon Ice Cap has suitable cross-track slopes (average slope ~0.7° - 1.5° over a distance of >2 km) for swath mode height estimation for both ascending and descending passes, and this area was used in the demonstration of swath mode processing (Gray et al. 2013). Figure 1 shows the positions of the spring 2011 surface height reference data

obtained from NASA and ESA supported overflights, and from surface skidoo dGPS transects, all superimposed on a colour representation of the reference DEM. The sub-satellite tracks of 15 CryoSat passes are also shown. Results from all the passes in this time period were compared to the reference surface heights as conditions on Devon Ice Cap change little between January and May, and we assume that any change in surface height or change in the bias between the surface and CryoSat height is small with respect to the error in the CryoSat heights.

The histogram of the difference between the reference and CryoSat swath mode heights obtained with the pre-launch baseline estimate (1.1676 m, Bouzinac, 2012) showed a bimodal distribution (Fig. 5C) and the average bias changed between ascending and descending passes. As we could find no reasonable geophysical explanation for this difference the possibility of a roll angle bias was investigated. If there is a roll angle bias on an ascending pass the swath-processed height estimates will be displaced either up- or down-slope depending on the sense of the bias. However, with a descending pass

and the same roll angle bias, the results will be displaced in the opposite direction and the height bias will have the opposite sign from that obtained with the ascending pass. To investigate this effect further all the data in this time period was reprocessed with an additional roll angle bias added to the value provided in the L1b file. Figure 4 illustrates the results of an experiment in which the 15 2011 passes (7 ascending and 8 descending) are each reprocessed 9 times with an additional roll correction varying from -0.02° to +0.02°. The results are then compared to the reference height data collected in early May

2011. As expected, the sense of the height difference changes between ascending and descending passes but the curves do not overlap well.  While the results from the 8 2011 descending passes do cluster nicely this is not the case for the 2012 data (Gray et al. 2016), and neither year shows consistent results for the ascending pass results. The satellite was in Earth shadow for the first 7 2011 descending passes over the test site and the sun elevation angle for the 8[th] pass on May 21 was only 4.7°, as it had just come out of earth shadow. However, the sun elevation angle history at the satellite for the 2011 ascending

passes and all the passes in 2012 was more variable, implying that plate bending due to solar heating is implicated in the roll angle problem.

Consequently, it appears that the roll angle provided in the L1b file has a time variable bias, presumably due to a variable





bending of the bench supporting the star trackers as the thermal environment changes. The uncertainty in the roll angle in this example appears to be of order 0.006° or ~ 100 µradians, not inconsistent with the observations in Galin (2013). While there will be a contribution from the range ambiguous zone in swath mode processing, which could introduce a small bias, this does not appear to be the primary source of these differences. The roll angle uncertainty, and resulting unknown bias in

the baseline roll angle, appears to be a limitation to the use of swath mode heights. Note that in Fig. 4 there is essentially no slope to the plots of the height difference versus roll angle bias for the POCA height estimates. This is direct consequence of the fact that while the POCA estimates are mapped incorrectly when there is a roll angle error, the derived height can still be appropriate for the wrong position because the incident wave may still be essentially perpendicular to the surface (Gray et al., 2013).

The variable E-W cross-track slope also provides a suitable test area to check the phase to cross-track angle conversion dictated by the baseline (Eq. 2 above). Figure 5 illustrates the results of an experiment in which the results obtained with a phase-to-angle conversion based on the pre-launch baseline are compared to the calibration given by Galin et al. (2013). The two histograms on the left used the pre-launch baseline while the histograms on the right used the angle scaling from Galin et al. (2013). Figure 5C shows the bimodal distribution referred to earlier, and Fig. 5A shows the improved results with a

significantly narrower error distribution when a bias of 0.0075° is subtracted from the roll angle provided in the L1b file. When the phase-to-angle is scaled by 0.973 (Fig. 5B and 5D) the results show a broader distribution and poorer results.

**3.3 Calibration based on data from west Greenland**

We use IceBridge data from an area in central western Greenland (Fig. 2, insert) including the Jakobshavn Glacier, an area which has shown significant surface height loss in recent years due to both change in output flux and surface mass balance

(Joughin et al., 2008, Qi and Braun, 2013), and has excellent reference surface height data (Krabill et al. 2002, Krabill 2014).

Figure 2 illustrates the positions of the reference surface height data obtained from the four NASA IceBridge flights flown on Mar. 31, and Apr. 6, 7, and 23 2011 superimposed on a black and white representation of the GIMP DEM (Howat et al. 2014). This DEM was used as the reference DEM for all the CryoSat processing in this area. Data from the ATM L2 files have been used for this work and compared with height results from all the CryoSat passes between mid-January and mid-

May.

It is important to recognize the differences in this test site in relation to that on Devon Ice Cap. The two profiles in Fig. 3 show that even in the accumulation area of this part of west Greenland the slope variation is much larger than on the EW profile interpolated from the airborne laser altimeter flown over of Devon Ice Cap. The difference is also very apparent in the CryoSat results: Figure 6 compares two image representations of the waveform power for 22 km segments of the Feb. 7

2011 ascending pass over Devon Ice Cap and the April 21 2011 descending pass over the west Greenland test site. For the ascending pass over Devon Ice Cap the POCA position will be on the left close to the beginning of the 240 m range window





as indicated by the stronger return signals in red. However, for the west Greenland site the peak return is often in the middle of the waveform. The difference in the signals may be influenced by the different conditions but it is clear that the dominant reason for the differences in waveforms is due to differences in the cross-track slopes. The larger slope variation in west Greenland clearly influence the CryoSat returns, and will adversely impact any retracker which uses all the waveform.

Figure 7 compares the results obtained with our geocoding and that obtained with CryoSat L2 retracker. Our processor picks out the POCA position satisfactorily (black dots on Fig. 7A) and leads to the mapping solution shown in Fig. 7B. The positions of the CryoSat L2 solutions are shown in Fig. 7B as red dots, and are often different by many kilometres. The solutions agree only when the waveforms show a clear maximum close to start of the waveform (e.g. at ~ 70.05 N). The histograms of the ATM minus CryoSat heights for all the 2011 spring data are shown in Fig. 8. Again the poor results from

the baseline C CryoSat L2 files are apparent (Fig 8A), particularly the much larger number of errors greater than 20 m. While it is unfair to compare results from an operational algorithm which must work everywhere to one which can be tuned for different areas and includes editing based on the coherence and the return power, it is fair to say that the west Greenland site appears to be inherently unsuitable for a retracker which uses the whole waveform. The L2 results are better in other areas, such as the ridges on Austfonna, ice rises in Antarctica and parts of the Devon Ice Cap, where the waveform shows a

more consistent shape and the dominant return is close to the start of the waveform.

The comparison between results obtained with the angle scaling factor from the Galin et al. (2013) calibration (Fig. 9A and C) and without (Fig. 9B and D) mirrors the results discussed in the previous section for Devon Ice Cap. The results imply that the pre-launch baseline coupled with an additional roll angle offset (or equivalent phase shift) improves the results for both west Greenland and Devon Ice Cap.

There is an important difference in the results for this test site in relation to Devon. For Devon the ATM - POCA height difference was essentially independent of the roll angle offset between -0.02° and 0.02° (Fig. 4), but this was not the case for the west Greenland site. A comparison of the average ATM - POCA height difference over 16 passes as a function of the additional roll angle bias (Fig. 10, top) shows that the CryoSat POCA height is not independent of the roll angle bias but increases for both positive and negative roll angle bias errors. As the CryoSat results are mapped incorrectly in the cross-

track direction the larger cross-track slopes imply that the distance in the cross-track direction which is essentially orthogonal to the incident wave is smaller in west Greenland than for the relatively smooth surface of western Devon Ice Cap. Consequently, this will lead to a CryoSat POCA height error as the mapping process takes the centre of the footprint outside the region which is orthogonal to the incident wave. Figure 10 (lower) shows the variation in the standard deviation of the swath mode ATM - CryoSat heights for each pass (dotted lines), and the average over all 16 passes (black line). The

offset in the position of the minimum from zero roll angle bias also supports the contention that on average there is a difference between the actual baseline roll angle and the value reported in the L1b file based on one of the 3 star trackers, or that there is an equivalent phase shift. For batch processing we have used the L1b roll angle with an additional roll angle bias of 0.0075°, but this may change with more experience with the bias and its variation with the time history of solar





illumination on the satellite.

There is another discrepancy in these results that warrants explanation. From Fig. 9 we see that the average ATM - POCA height difference is -0.16 m (Fig. 9A), but with the same waveform data the height difference from swath mode processing is +0.91 m (Fig. 9C), so that the two processing methods are giving average heights different by 1.07 m. With the Galin et al.

(2013) calibration the discrepancy is even worse; 2.52 m. Further, there is an apparent discrepancy with the results from Devon Ice Cap where previously (Gray et al. 2015), and now, we see the CryoSat height as being somewhat below the physical surface. The explanation for the anomalous average ATM – POCA result for west Greenland, where the average CryoSat POCA height is slightly above the surface, appears to be related to the results in Fig. 10A. If there is an error in the roll angle this will lead to an increase in detected height irrespective of the sign of the roll angle error. This will lead to an

asymmetric distribution and the mean height will be biased high. Note that the distribution in Fig. 9A is somewhat asymmetric, more so than that in Fig 9C for the swath processed data where the sign of any roll angle error would dictate the sign of the height error. For areas like the west Greenland test site this implies that the roll angle bias error will tend to bias the average POCA height high with respect to the surface.

## 4 Unique capabilities of the SARIn mode.

In this section we use our methodology and revised calibration to demonstrate some unique capabilities of the SARIn mode, first by illustrating signature change with surface conditions in West Greenland, and secondly by showing that it is possible to detect supraglacial lakes in the waveform data and estimate the surface height and height change with relatively good accuracy.

### 4.1 The effect of surface melt on SARIn waveforms

The influence of melt on SARIn signatures should be considered when presenting temporal height change for any region which may have undergone surface melt (Nilsson et al., 2015, Gray et al., 2015). Figure 11 illustrates one example of the influence of melt on the strength of the SARIn waveform data. The position of this July 14 2011 descending pass is shown in Fig. 12A and begins at ~ 2200 m elevation, crosses the Jakobshavn Glacier at ~ 1000 m, then the elevation increases slightly before ending at ~ 1100 m. At high elevations, the returns are comparable to those obtained under cold winter-spring

conditions, but at lower elevations, ~ 1700 - 1900 m, there is a decrease of ~15 - 20 dB in average waveform power. We speculate that this is due to the relatively low reflectivity and high absorption of a damp snow layer in which the moisture is distributed as small droplets. At lower elevations (< 1600 m) not only is the average return larger but also the waveform-to-waveform variability is much higher, indicative of occasional specular reflection from a wet surface facing the radar. Again, we speculate that at this stage the moisture has increased and coalesced to the point that there is occasional coherent

reinforcement of the return signal from a wet and therefore reflective surface. Also, the strongest returns in most of the





waveforms in this area are not from the leading edge but vary in position across the waveform so that a retracker that uses all of the waveform won't accurately measure the position and height of the POCA.

Figure 12 illustrates the average waveform power plotted against elevation for 5 descending passes (Fig. 12A) acquired during the summer of 2011. At elevations up to ~ 1300 m the June 18 pass (Fig. 12D) shows the high waveform-to-waveform variability that we speculate is due to occasional specular reflection, but this was not observed in the earlier passes in April (Fig. 12B) and May (Fig. 12C). By July 14 (Fig. 12 E) the region with strong and variable power includes elevations up to ~1600 m and the August pass (Fig. 12F) shows some strong waveform returns at even higher elevations. Comparable results were obtained from the five repeat passes 369 days later in 2012, but the descending pass on July 20 2013 showed the wet snow signature at lower elevations (~ 1500 m) without any indication of occasional specular reflections. This is consistent with the relatively colder conditions at that time in 2013 with respect to both 2011 and 2012 (see e.g. Fettweis 2016, http://climato.be/melt-2016). The supporting material includes figures equivalent to Fig. 12 for all the years from 2012 to 2016.

## 4.2 Supraglacial Lakes

During summer melt around the periphery of the Greenland Ice Sheet water pools in surface depressions as supraglacial lakes (Echelmeyer et al., 1991), forming first at lower elevations and then to higher elevations as melt progresses. With increasing positive air temperatures, surface melt water will infiltrate to lower elevations so that the snow at the edges of the depression will tend to become saturated and melt before the snow in the centre of the depression. In many cases, small supraglacial streams form which will add energy to melt snow or ice where they enter the surface depression.

Optical satellite imagery has been used to study the distribution, extent, depth and drainage of these features when there is an open water surface (Box and Ski, 2007, McMillan et al. 2007, Sneed and Hamilton 2007, Liang et al., 2012, Fitzpatrick et al. 2014, Pope et al., 2016). While Landsat and MODIS imagery have been used to estimate total lake volume of relatively large areas (e.g. Pope et al., 2016), the limitations due to clouds and atmospheric conditions hamper routine use for quantitative melt estimates. Here we demonstrate that CryoSat SARIn data can provide complementary information to that available from visible satellites by showing that measurements of surface height and height change can be derived from SARIn data over individual supraglacial lakes. SARIn data can be obtained reliably day or night and in all weather conditions, but is very limited in surface coverage.

If CryoSat passes directly over a typical unfrozen supraglacial lake one would expect a strong specular reflection which would not be at the leading edge of the waveform, as it must be surrounded by ice at higher elevations. Even if the lake has some snow cover or a partially unfrozen surface, the flat surface will still enhance the return and could lead to a strong peak in the waveform. Figure 13 illustrates some strong signals in the middle of the waveforms of a 50 km section of the 7 August 2011 ascending pass over the test area in west Greenland. These may originate from extended surfaces orthogonal, or nearly



orthogonal, to the incident wave. We have selected one such strong signal, labelled as 'L1' in Fig. 13, which is detected in results from ascending and descending passes from all the summers from 2010 to 2016. The supplementary material contains a sequence of 14 summer MODIS images from 2012 to 2016 which show that the L1 and L2 features are above the snow line for all 5 years and that the surface of these depressions did not become totally ice free. Figure 14 shows the positions of

the sub-satellite tracks superimposed on a summer Landsat 8 image and that there were dark regions, presumably wet snow, at the positions of the topographic lows marked as L1 and L2. The relative strength of the CryoSat return signals for the six ascending passes for both features are shown in Fig. 15 and the year-to-year derived height in Fig. 16 with details provided in Table 1. The sequence of dates for the repeat ascending passes are Aug. 4, 2010; Aug. 7, 2011; Aug. 9, 2012; Aug. 12, 2013; Aug 16, 2014; Aug. 19, 2015, and Aug. 21, 2016 reflecting the 369.25-day repeat orbit cycle. The repeat descending

passes are 5.5 days later.

Our interpretation of the strengths of the lake signatures and the surface elevation is as follows: Considering the low surface velocity (~ 3.5 m/year, Joughin et al., 2012, 2016) and elevation (~1600 m) at this position, it is unlikely that either of these depressions drained in any of the summers. The increase in height from the summer 2010 to 2012 (Fig. 16) appears to reflect the increase in melt at this position, particularly in 2012 (see e.g. Fettweis, http://climato.be/melt-2016). The specific causes

of the subsequent decrease in elevation are not known.

The Landsat 8 image from July 6 2016 (Fig. 17) includes one 2.4 x 1 km lake at 70.37 N, 49.79 W, and ~1020 m in elevation, which was detected in the CryoSat waveforms from all the ascending and descending repeat passes listed on Fig. 17 between 2011 and 2016. By the time of the Landsat 8 image in 2016 most of the snow had melted and we surmise that melt had been on-going during June and early July for the years 2011 – 2016 at this position, and at the times of the CryoSat

over-passes (Fig. 17 and Table 2). Figure 18 illustrates the lake height for all passes except for the 2013 descending pass which was too far to the west of the lake for reliable results. In contrast to the high elevation, low melt 'lake' described above, now there is a clear height increase in the 5.5 days between the ascending and descending passes over the lake. This allows an estimate of the filling rate at the time of the two passes. If we assume a lake area of $2 \pm 0.5$ km$^2$ (~0.2. $10^6$ – 2.$10^6$ m$^3$ melt water added per day). This lake does drain sometime after the start of July, see the MODIS sequence in the

supplementary material, but appears not to have drained at the times of any of the CryoSat overpasses.

**5 Discussion**

In this section we discuss the two SARIn processing approaches, the limitations and successes of the current CryoSat SARIn products for glacial ice, and then speculate on some of the characteristics of a future interferometric radar altimeter for monitoring ice cap change.

There are two important advantages with swath processing: firstly, there is no need for a retracker and, secondly, the swath data is obtained predominantly from the region directly beneath the satellite and the look angles for the swath footprints can



be less than those for the POCA (for those areas with cross-track slopes appropriate for swath processing). With the small look angles, the footprint illumination cross-track is essentially uniform. Consequently, assuming a small contribution from the range ambiguous area, the phase should represent the geometric centre of the footprint so that the range, satellite state vectors, and the various angles should lead to reliable heights. Unfortunately, the roll angle problem discussed earlier

compromises the swath mode results as the resulting cross-track mis-mapping will normally lead to a height error (Gray et al., 2013).

POCA processing requires a retracker and the look angle can extend into the range in which the illumination cross-track is affected by the antenna pattern variation so that the phase may not reflect the geometric centre of the footprint. Rather it would be displaced towards the sub-satellite track. With interferometric swath processing precise knowledge of the baseline

and baseline angles is important (Rosen et al. 2000), and with the CryoSat roll angle problem POCA heights are normally better suited to temporal height change estimation than swath mode heights. Consequently, while the average swath mode bias to the surface height may be more realistic than the POCA results under some conditions, height change results are normally better based on CryoSat POCA data.

The known problem of bending in the star-tracker support plate and the resulting varying error in the reported value of the

baseline roll angle does have an impact on the accuracy of the CryoSat height results. Any roll angle error translates directly into a cross-track mapping error so that the resulting height error then depends on the angle between the incident wave and the tangent to the cross-track surface. If this angle is 90° and the surface slope changes slowly over a few hundred meters, then the error is very small as the geocoding algorithm produces the correct elevation for the mis-mapped footprint. Although we show that the roll angle problem had essentially no impact on the Devon POCA results, it did have an impact

on the POCA results from the west Greenland test site. In this case the cross-track slopes varied more rapidly than for Devon and lead to the situation where an incorrect roll angle could lead to an increase in the CryoSat height with respect to the surface irrespective of the sense of the roll angle error. This we suggest is the origin of the unrealistic result that the average POCA height can be slightly above the physical surface for the west Greenland site.

POCA heights originate from ridges and peaks and, when the cross-track slope is appropriate for swath processing, the swath

mode results will normally originate from the area beneath the satellite so the two approaches are complementary in surface coverage. As discussed above, there can be a bias between POCA and swath heights which needs to be considered if the results are merged. The potential height error for individual estimates is normally less for POCA data than for swath mode heights but the exception is the accuracy with which one can estimate the height of relatively large supraglacial lakes when the lake is beneath the satellite and viewed at close to normal incidence. In this case, we have a very strong signal in the

middle of the waveform, any range ambiguous contribution should be small, and no retracker is required for the geocoding solution. Further, with this viewing geometry the problem of an incorrect roll angle leads to a small error in the lake surface height and accuracies of ~ 0.5 m are possible for the surface height of a large lake. Work is underway to better evaluate the extent to which CryoSat data can help in quantifying the time and extent of melt around Greenland.



The ability to geocode the relatively small footprint possible with the SARIn mode over glacial ice creates a huge advantage for this mode over the traditional low resolution radar altimetry. Future radar altimeters employing coherent along-track processing, either fully focussed or Delay-Doppler, coupled with cross-track interferometry, could play a very important role in monitoring change on many ice caps and glaciers. However, a sun synchronous orbit, preferably dawn-dusk to minimize the impact of changing solar illumination on the interferometric baseline, could improve the results. As users are primarily concerned with change year-to-year a 73 or 365-day repeat cycle would also be ideal, if possible.

## 6 Conclusions

Here we list the specific conclusions arising from our analysis of the SARIn data over Devon Ice Cap and west Greenland.

1. A more consistent fit can be obtained between CryoSat and surface heights using the prelaunch baseline coupled with an additional roll angle bias of ~ 0.0075°. Although the additional bias may originate with the angle measurement, it could equally well be an equivalent, additional phase correction of ~0.0435 radians to the value of 0.612 radians currently used in the baseline C product (Bouzinac, 2012).
2. A retracker which uses the first significant leading edge of the waveform normally leads to more reliable elevations than a retracker that uses the whole waveform, this appears to be particularly true for areas like West Greenland in which the shape of the waveform is very variable and the peak signal is often in the middle of the waveform.
3. Swath mode results complement the POCA results but are normally less accurate. The exception is the accuracy with which the heights of supraglacial lakes can be obtained when the satellite flies almost directly over the lake.
4. The uncertainty in the CryoSat baseline roll angle affects primarily swath mode results but can also impact the accuracy of POCA results when the surface topography is comparable to that in the west Greenland test site.
5. While more work is required to establish to what extent CryoSat SARIn waveforms and heights can improve our knowledge of melt in the ablation zone of the Greenland Ice Sheet, these initial results are encouraging that CryoSat SARIn data can help provide useful information on the variation of year-to-year melt.

## Acknowledgements

This work was supported by the European Space Agency through the provision of CryoSat-2 data and the support for the CRYOVEX airborne field campaigns in both the Canadian Arctic and Greenland. NASA supported the IceBridge flights over the Canadian Arctic and Greenland, while NSIDC facilitated provision of the airborne laser data. The Technical University of Denmark (TUD) managed the ESA supported flights over Devon. The IceBridge and TUD teams are gratefully acknowledged for the acquisition and provision of the airborne data used in this work. The Polar Continental Shelf Project





(Natural Resources Canada) provided logistic support for field work in the Canadian Arctic, and the Nunavut Research Institute and the community of Resolute Bay gave permission to conduct research on the Devon Ice Cap. Support for D. Burgess was provided through the Climate Change Geoscience Program, Earth Sciences Sector, Natural Resources Canada and the GRIP programme of the Canadian Space Agency. Support for K. Langley was provided by ESA project Glaciers-CCI (4000109873/14/I-NB). T. Dunse and G. Moholdt were supported by ESA-Prodex project 4000 110 725/724 "CRYOVEX" and T. Dunse was supported by the Nordforsk-funded project Green Growth Based on Marine Resources: Ecological and Sociological Economic Constraints (GreenMAR). Wesley Van Wychen and Tyler de Jong helped with the 2011 kinematic GPS survey on Devon. NSERC funding to L. Copland is gratefully acknowledged. We also acknowledge NASA and NSIDC for the provision of the Landsat 8 and MODIS imagery.



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




| Year | Date | Pass direction | Local time | Sun elevation angle (deg.) | Min dB | Standard deviation, m (no. of samples) | Look angle (deg.) | Height (m) | Height error (estimated, ± m) |
|---|---|---|---|---|---|---|---|---|---|
| L1 | | | | | | | | | |
| 2010 | Aug 4 | Ascending | 1:03 | 0 | -130 | 0.52 (61) | 0.08±0.006° | 1606.5 | 0.5 |
| | Aug 9 | Descending | 13:33 | 36 | -135 | 0.58 (61) | -0.18±0.006° | 1606.3 | 0.6 |
| 2011 | Aug 7 | Ascending | 6:48 | 14 | -130 | 0.48 (51) | -0.05±0.006° | 1609.1 | 0.5 |
| | Aug 12 | Descending | 19:18 | 14 | -134 | 0.41 (57) | -0.22±0.006° | 1609.5 | 0.5 |
| 2012 | Aug 9 | Ascending | 12:33 | 31 | -125 | 0.63 (52) | -0.05±0.006° | 1613.8 | 0.6 |
| | Aug 15 | Descending | 1:03 | -7 | -127 | 0.29 (59) | -0.2±0.006° | 1614.3 | 0.4 |
| 2013 | Aug 12 | Ascending | 18:16 | 19 | -135 | 0.59 (73) | -0.1±0.006° | 1612.5 | 0.6 |
| | Aug 18 | Descending | 6:46 | 9 | -139 | 0.33 (48) | -0.2±0.006° | 1612.8 | 0.4 |
| 2014 | Aug 16 | Ascending | 0:01 | -2 | -138 | 0.56 (75) | 0.05±0.006° | 1610.8 | 0.6 |
| | Aug 21 | Descending | 12:32 | 32 | -130 | 0.46 (74) | -0.18±0.006° | 1611.2 | 0.5 |
| 2015 | Aug 19 | Ascending | 5:46 | 6 | -138 | 0.64 (48) | -0.04±0.006° | 1610.2 | 0.6 |
| | Aug 24 | Descending | 18:16 | 16 | -139 | 0.45 (34) | -0.15±0.006° | 1610.3 | 0.5 |
| 2016 | Aug. 21 | Ascending | 11:31 | 27 | -140 | 0.49 (34) | -0.02±0.006° | 1606.1 | 0.5 |
| | Aug. 27 | Descending | 0:01 | -14 | -140 | 0.51 (18) | -0.17±0.006° | 1606.7 | 0.6 |
| L2 | | | | | | | | | |
| 2010 | Aug 4 | Ascending | 1:03 | 0 | -120 | 0.75 (41) | 0.01±0.006° | 1571.5 | 0.7 |
| 2011 | Aug 7 | Ascending | 6:48 | 14 | -134 | 0.42 (57) | -0.14±0.006° | 1573.1 | 0.5 |
| 2012 | Aug 9 | Ascending | 12:33 | 31 | -130 | 0.35 (54) | -0.1±0.006° | 1576.0 | 0.4 |
| 2013 | Aug 12 | Ascending | 18:16 | 19 | -135 | 0.32 (16) | -0.16±0.006° | 1573.2 | 0.5 |
| 2014 | Aug 15 | Ascending | 0:01 | -2 | -135 | 0.37 (30) | -0.03±0.006° | 1572.1 | 0.4 |
| 2015 | Aug 19 | Ascending | 5:46 | 6 | -137 | 0.43 (32) | -0.13±0.006° | 1572.5 | 0.5 |
| 2016 | Aug. 21 | Ascending | 11:31 | 27 | -133 | 0.47 (27) | -0.10±0.006° | 1573.1 | 0.5 |

Table 1.

5  Information on the conditions and results of the analysis of the CryoSat data for the two lake features L1 (70.275 N, 48.56 W) and L2 (70.178 N, 48.55 W) shown in Figs 13 and 14. The 'Min dB' column reflects the lower limit of the sample power used in the averaging of the height estimates contained within the window around the surface depression.

| Year | Date | Pass direction | Local time | Sun elevation angle (deg.) | Min dB | SD height (no of samples) | Look angle (deg.) | Height (m) | Height error (estimated ± m) |
|---|---|---|---|---|---|---|---|---|---|
| 2011 | June 14 | ascending | 9:33 | 31 | -125 | 0.74 (38) | 0.03±0.006° | 1014.7 | 0.7 |
| | June 20 | descending | 22:03 | 9 | -130 | 0.59 (41) | -0.07±0.006° | 1016.8 | 0.6 |
| 2012 | June 16 | ascending | 15:18 | 37 | -115 | 0.77 (56) | 0.003±0.006° | 1020.4 | 0.8 |
| | June 22 | descending | 3:48 | 6 | -120 | 0.80 (45) | -0.01±0.006° | 1022.3 | 0.8 |
| 2013 | June 19 | ascending | 21:02 | 15 | -130 | 0.57 (48) | -0.02±0.006° | 1017.1 | 0.6 |
| | June 25 | descending | | | | | | | |
| 2014 | June 23 | ascending | 2:46 | 8 | -130 | 0.37 (23) | 0.06±0.006° | 1018.7 | 0.4 |
| | June 28 | descending | 15:16 | 41 | -135 | 0.53 (22) | 0.03±0.006° | 1020.2 | 0.5 |
| 2015 | June 26 | ascending | 8:31 | 27 | -130 | 0.67 (39) | -0.001±0.006° | 1017.6 | 0.7 |
| | July 1 | descending | 21:01 | 13 | -130 | 0.87 (29) | -0.07±0.006° | 1020.1 | 0.9 |
| 2016 | June 28 | ascending | 14:16 | 43 | -130 | 0.74 (62) | 0.03±0.006° | 1021.4 | 0.7 |
| | July 4 | descending | 2;46 | -1 | -130 | 0.64 (29) | 0.02±0.006° | 1022.1 | 0.6 |

Table 2.
Information on the conditions and results of the analysis of the CryoSat data for the lake shown in Figs 17 and over-flown by CryoSat on the dates shown.





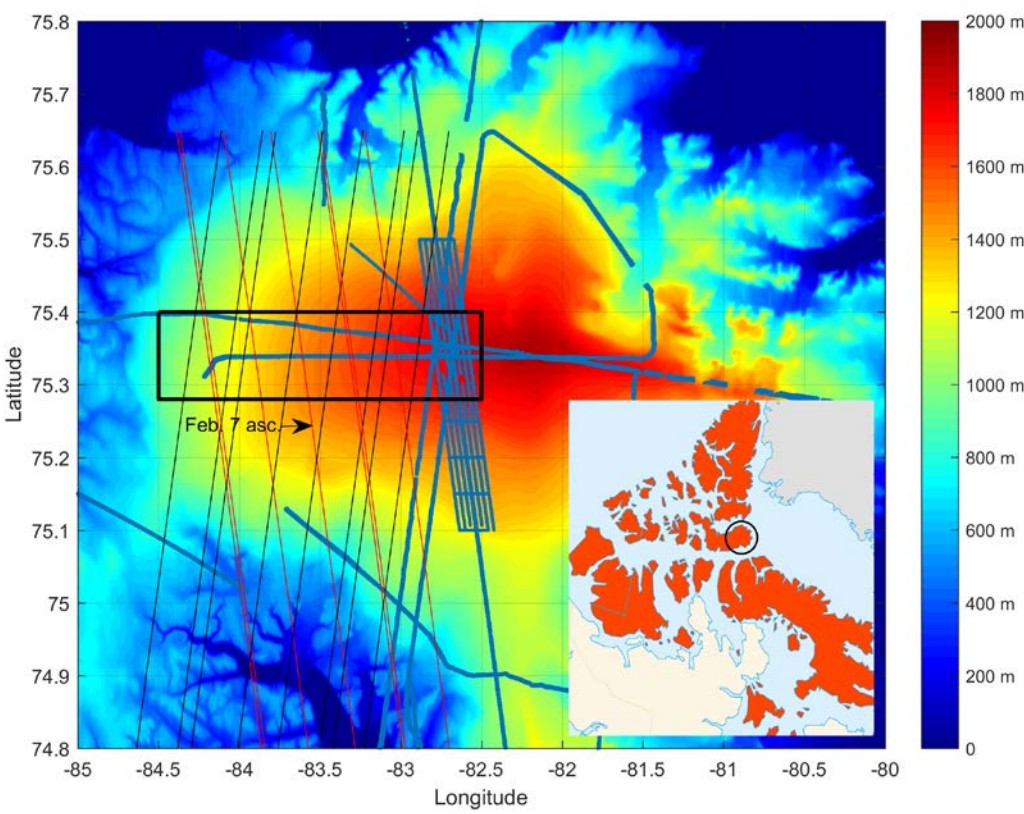

5   **Figure 1: Location of the test area on the western slopes of the Devon Ice Cap (black rectangle). The sub-satellite positions of the spring 2011 ascending and descending passes crossing the test area are shown by the red and black lines respectively. The positions of the reference surface height data are shown in blue. The insert shows the position of Devon Ice Cap (circled) in the Canadian Arctic Archipelago.**





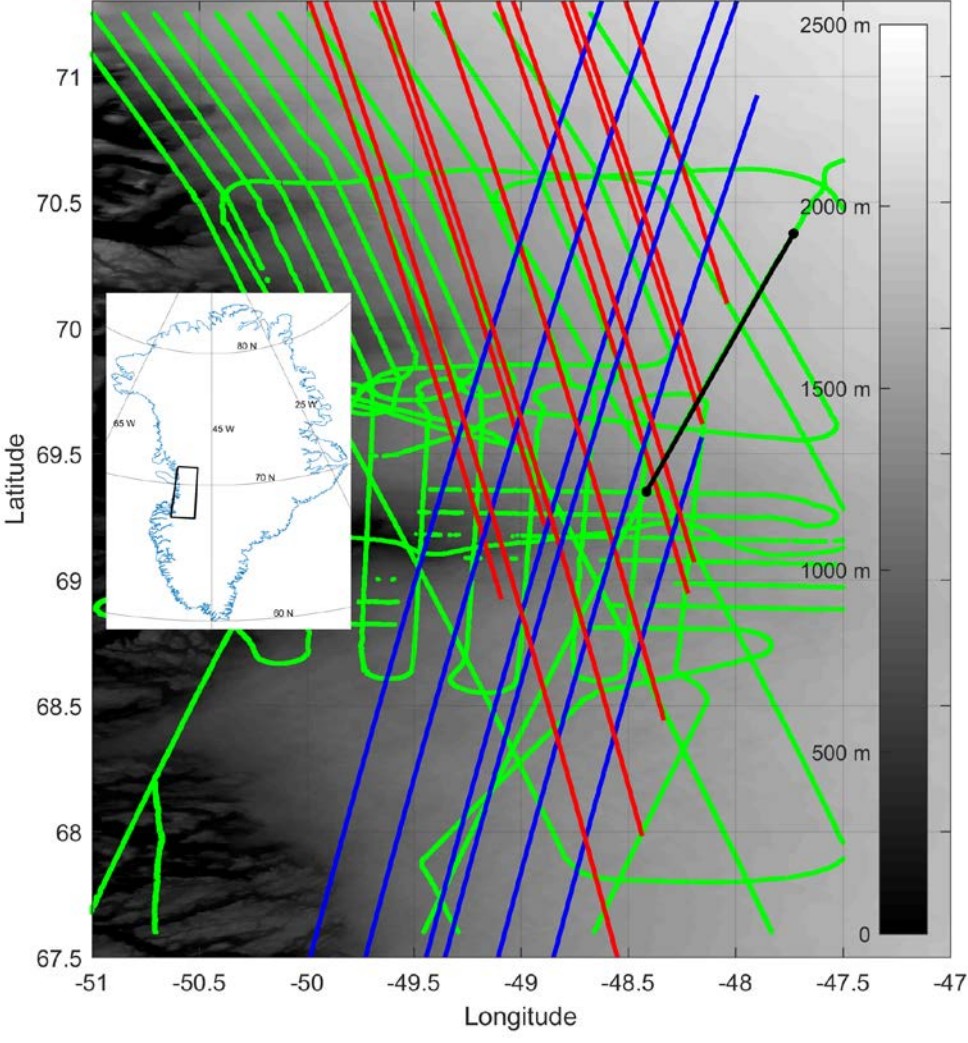

**Figure 2: The positions of the reference ATM surface elevations flown by NASA IceBridge missions over the west Greenland site in spring 2011 are shown in green. Sub-satellite CryoSat tracks for the period Jan. 20 to May 16 2011 are shown by red (ascending) and blue (descending) lines. The inset map shows the position of the test area in Greenland and the background image is a black-white representation of the GIMP reference DEM (Howat et al., 2014). The position of the height profile in Fig. 3A is shown by the black line.**





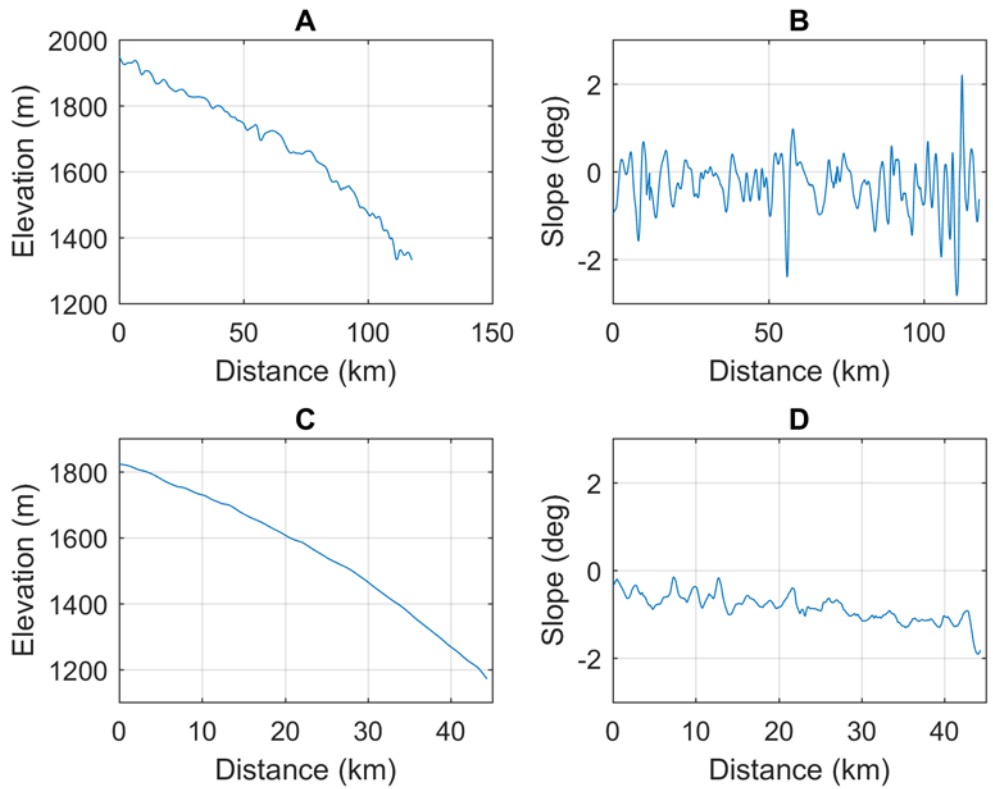

**Figure 3: Illustration of the difference in slopes for a typical west Greenland transect (70.37°N, -47.73°W to 69.35°N, -48.42°W, black line in Fig. 2) derived from an ATM flight line from Apr. 6 2011 (box A; elevation and box B; slope) and the EW transect from western Devon Ice Cap (box C; elevation and box D slope, Fig. 1).**





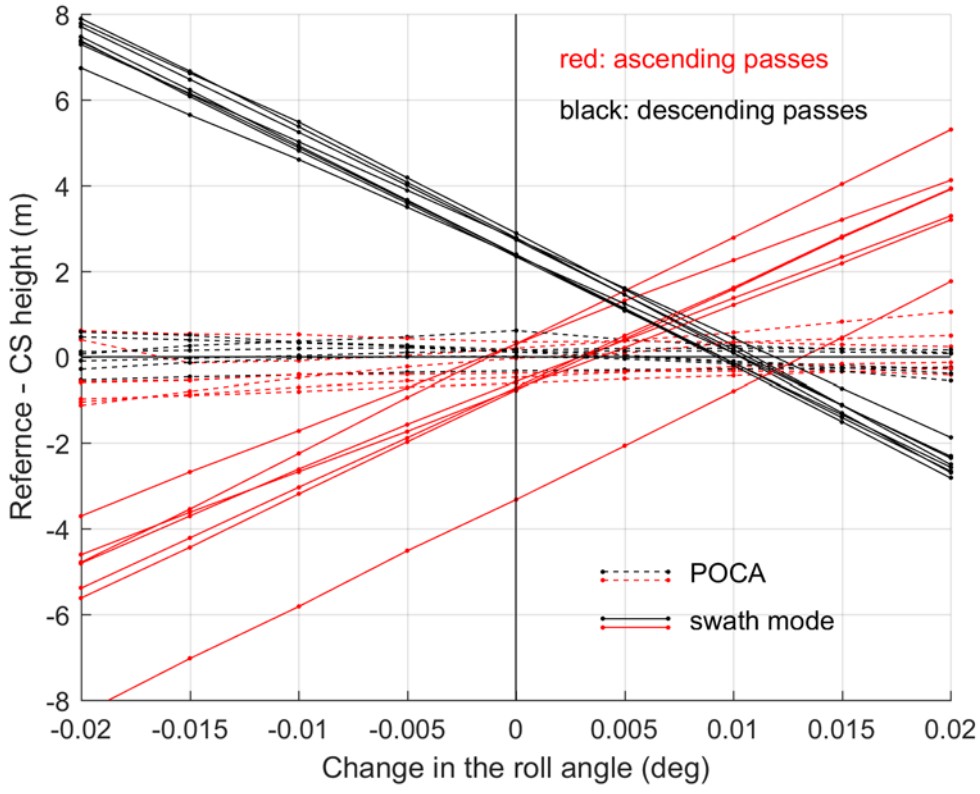

**Figure 4: Illustration of the changing bias between the reference and CryoSat (CS) swath mode heights for the Devon test site as an additional roll bias is subtracted from the roll figure given in the L1b file. Results for 7 ascending and 8 descending passes in the winter-spring of 2011 are shown in red and black, respectively. The reference - POCA height variation with the added roll angle bias is shown with the dashed lines.**





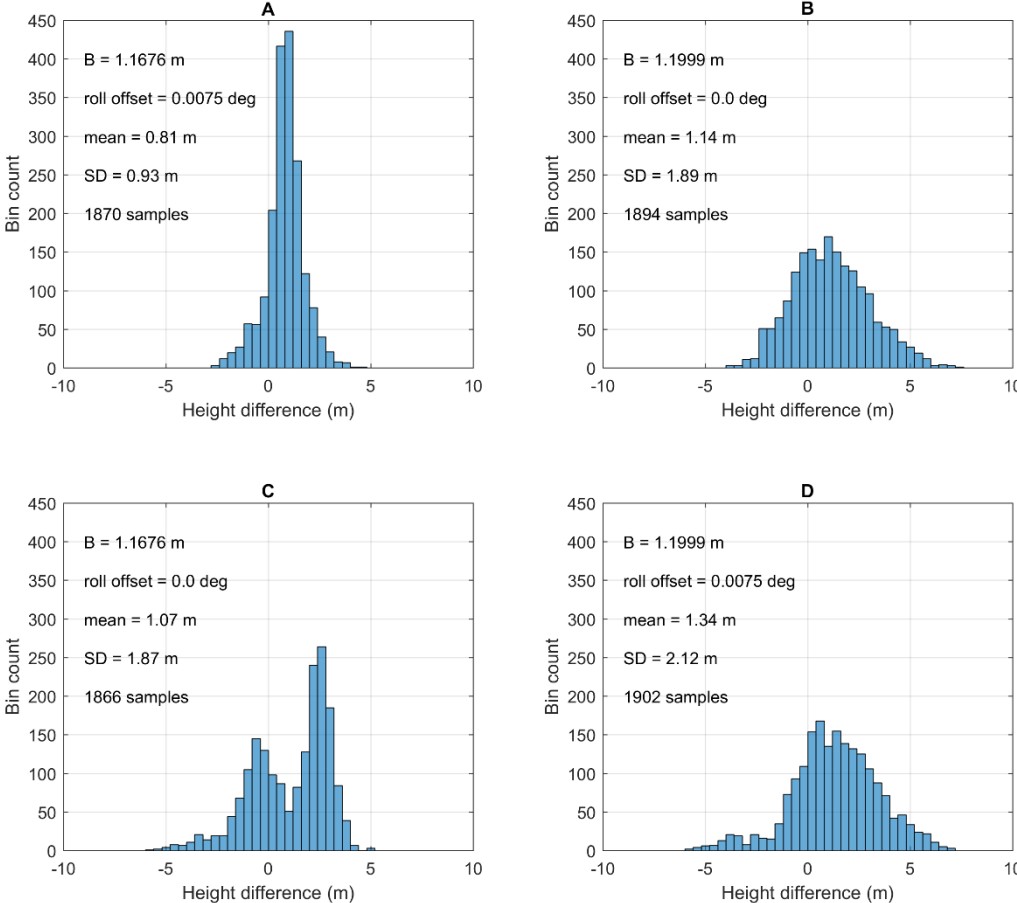

**Figure 5: Histograms of the reference minus CryoSat swath heights for Devon Ice Cap: (A) pre-launch baseline and a roll angle offset of 0.0075°; (B) modified baseline with zero roll offset; (C) pre-launch baseline with zero roll offset; and (D) modified baseline with a roll offset of 0.0075°.**





**Figure 6. Waveform power for 22 km segments of (A) the Feb. 7 2011 ascending pass over Devon Ice Cap (Fig. 1) and (B) the April 21 2011 descending pass (Fig. 12A) over the west Greenland test site. The return power in dB is represented in colour and the individual waveforms have been shifted in the x direction depending on the time delay to the first sample and the satellite elevation above the WGS84 ellipsoid.**



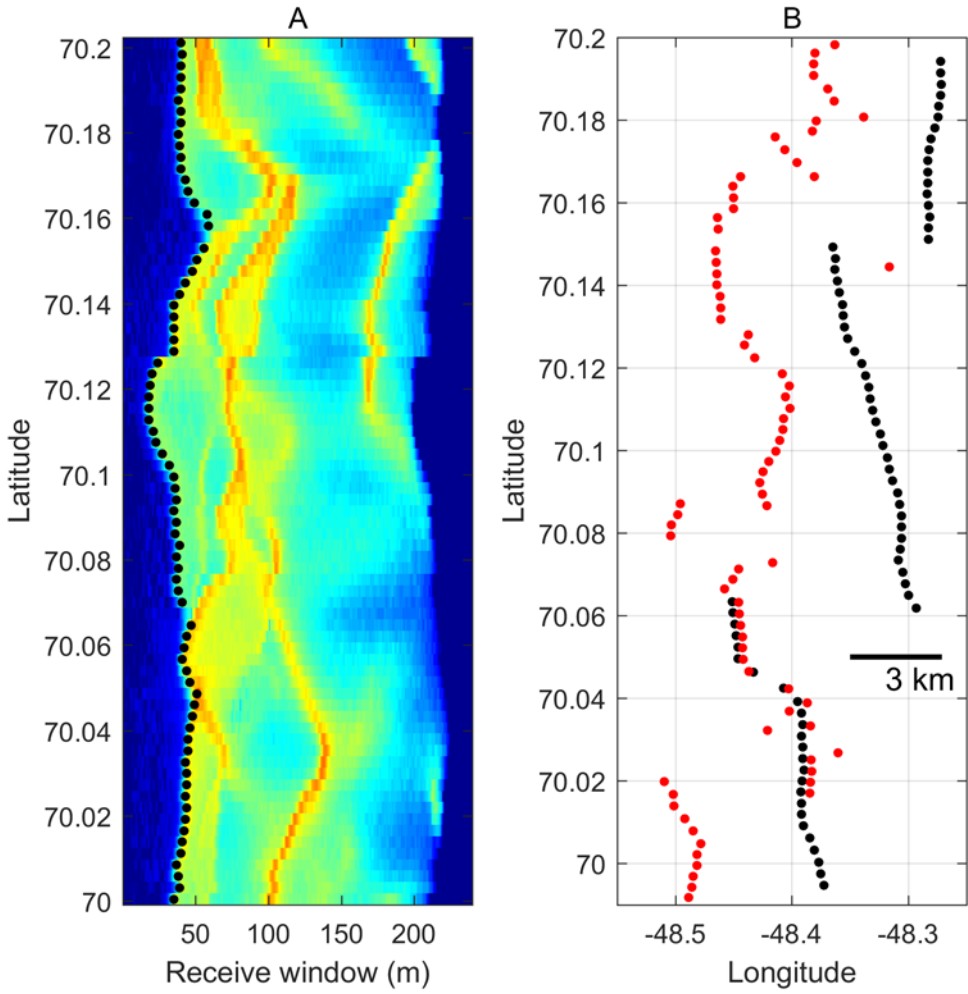

**Figure 7. (A) Waveform power without any x axis shifts using the same dB colour scale as in Fig. 6B. The detected POCA positions are shown in (A) for each waveform with black dots, and they clearly correspond to the leading edge of the waveforms. (B) Geographic positions of the geocoded footprints (black dots) are compared to the positions of the ESA L2 solutions (red dots).**



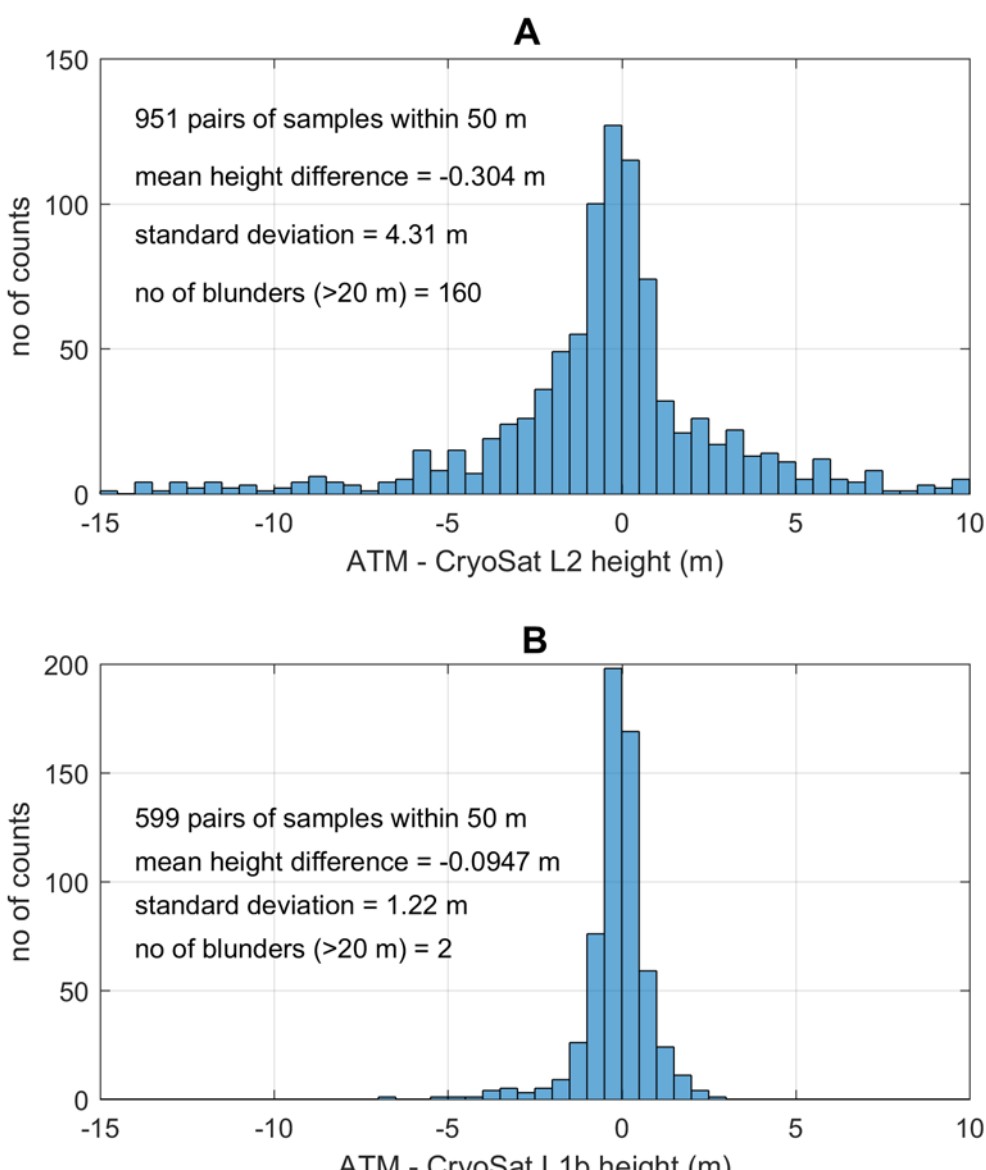

**Figure 8. Comparisons of the ATM - CryoSat POCA height difference histograms for the west Greenland test site. (A) the ESA L2 solution: (B) Results from the current maximum slope leading edge retracker.**





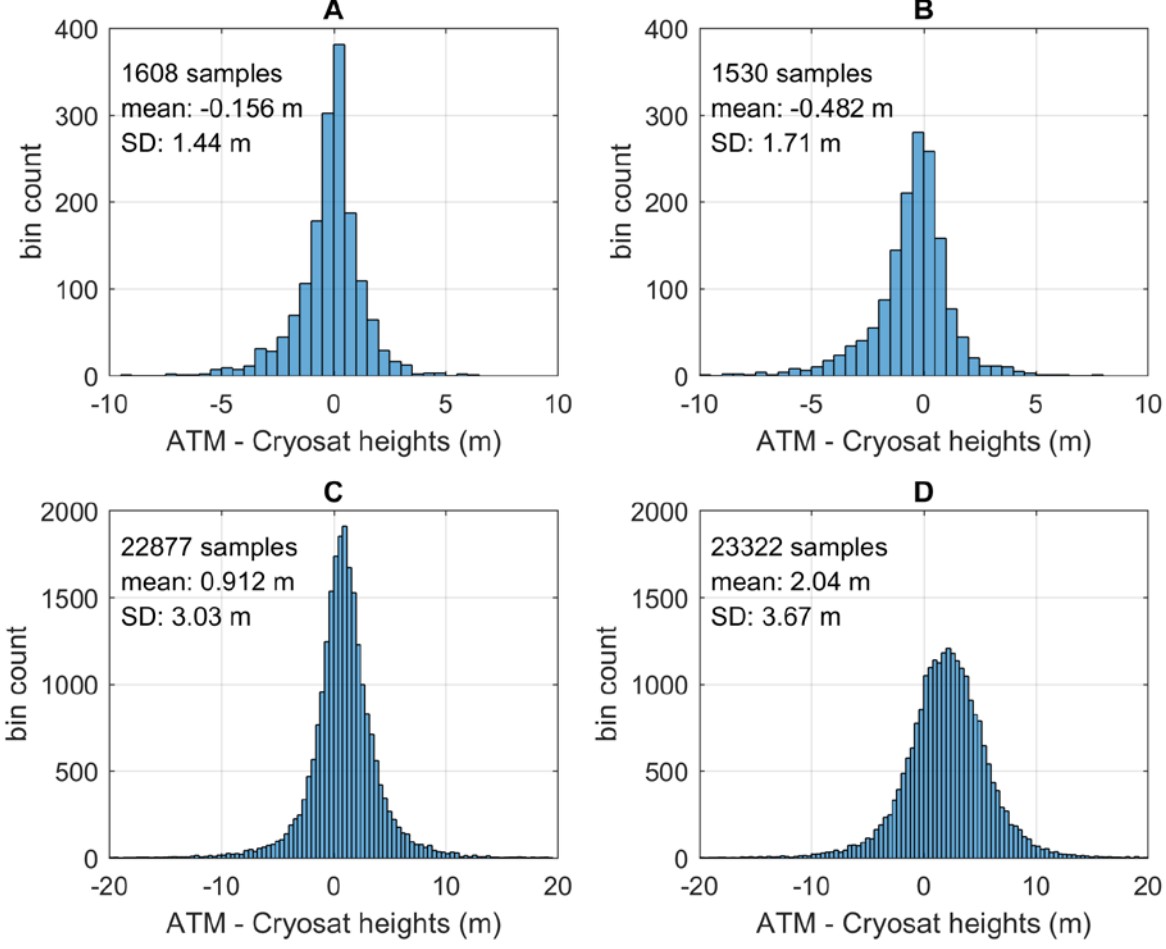

**Figure 9. Comparisons of the west Greenland ATM – CryoSat height difference histograms for the solution using the pre-launch baseline coupled with an additional roll offset (A; POCA, C; swath mode solutions). Boxes B and D use the Galin et al. (2013) calibration for the POCA and swath solutions respectively.**





**Figure 10. Illustration of the average of the ATM – CryoSat height differences for: (A) 16 passes plotted against the additional roll angle bias used in the processing (top). The error bars are ± 1 standard deviation about the mean. (B) Variation in the standard deviation (SD) of the ATM – CryoSat height difference for the individual passes (blue dotted lines) and the average over all the passes (solid black line).**



**Figure 11. The background image illustrates the swath waveform power in colour with a dB scale for the July 14 2011 descending pass over the west Greenland test area (Fig. 12A). The waveforms making up this pseudo-image have been shifted in the x direction to account for the changing delay time to the first sample and the varying satellite height above the WGS84 ellipsoid. The insert shows the sub-satellite terrain elevation and the waveform average power both plotted against latitude.**





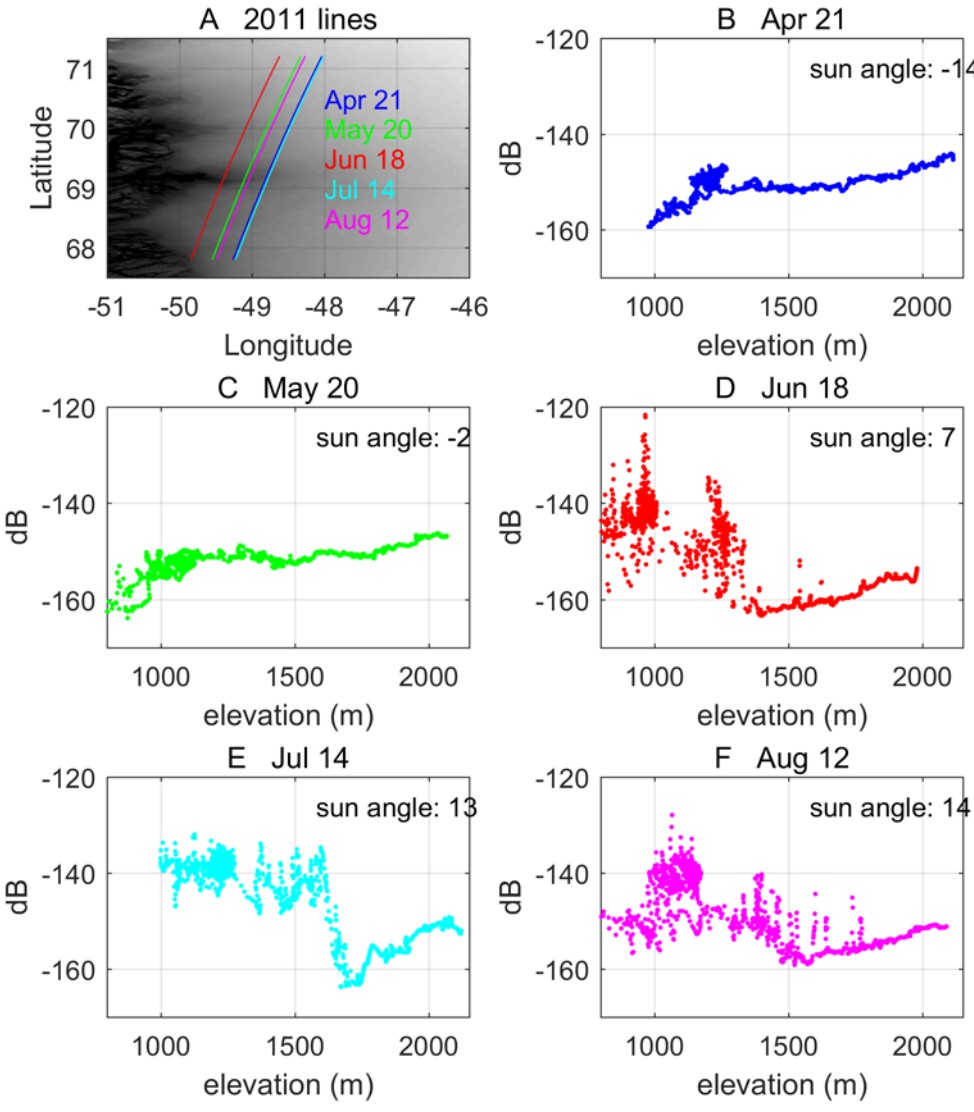

**Figure 12.** Plots of the average waveform power for the five 2011 descending passes shown in (A). The five plots are from descending passes on (B) April 21, (C) May 20, (D) June 18, (E) July 14 and (F) Aug. 12, and illustrate average waveform power as a function of elevation.




**Figure 13. Illustration of part of the waveform power from an ascending pass over west Greenland on 7 August 2011**
5 **(Fig. 14). The bright returns labelled as L1 and L2 are at elevations ~ 1609 m and 1573 m, respectively, and represent**
**topographic lows where water could collect.**





**Figure 14. Ascending and descending sub-satellite repeat tracks over, or close to, the L1 and L2 features for all the years from 2010 to 2016 superimposed on part of the Landsat 8 image of August 9 2016 (inset image).**





**Figure 15. 'Images' of part of the CryoSat waveforms for the areas including 'L1' (top) and 'L2' (bottom) in west Greenland for the August dates in each year from 2010 to 2016.**





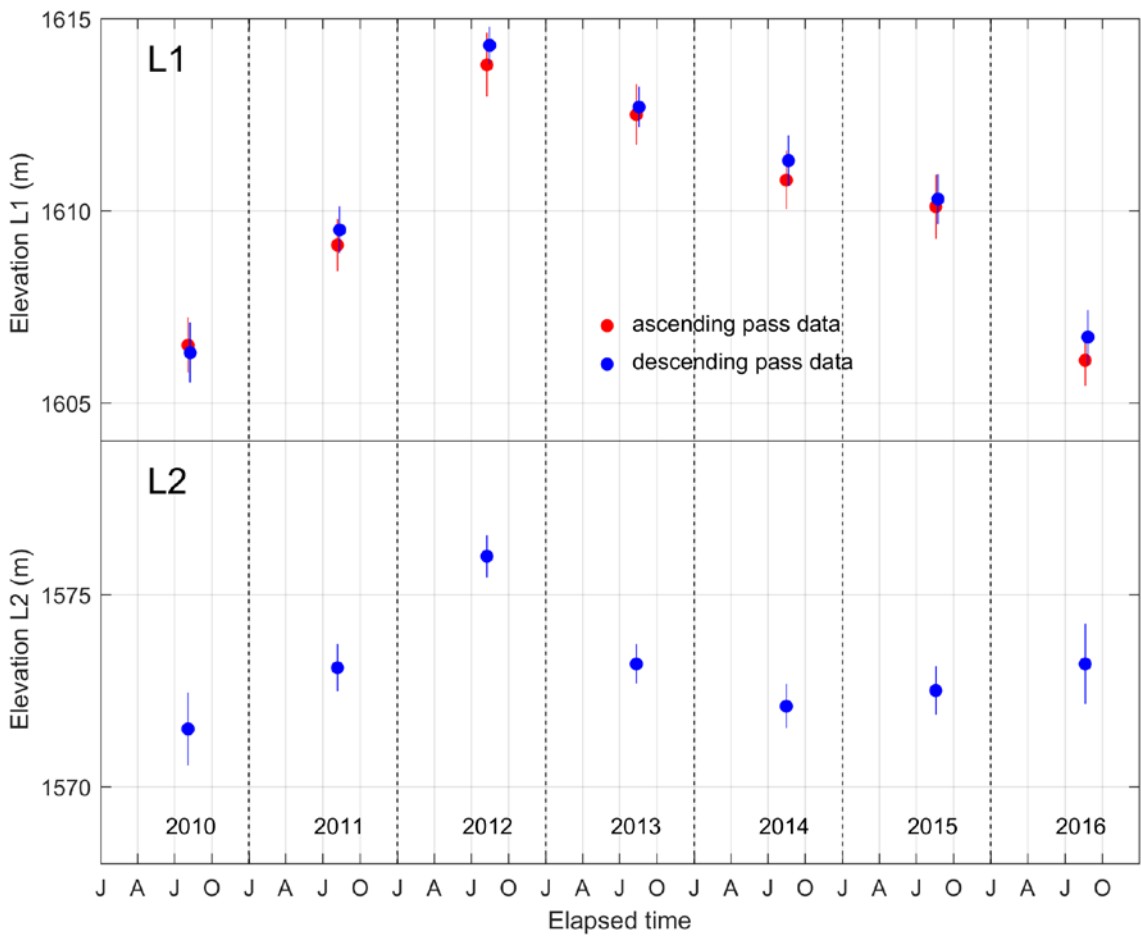

**Figure 16. Surface elevation of L1 (top) and L2 (bottom) between the summers of 2010 and 2016.**





**Figure 17. Landsat 8 image from July 6 2016 of an area in the ablation zone of the west Greenland test site which includes a lake (white arrow) viewed by CryoSat on all the repeat ascending and descending passes listed on the image. The insert image shows the blow-up position in the full Landsat 8 frame.**



**Figure 18. (A) Surface height of the lake in Fig. 17 at the times of the overpasses: (B) Height increase during the 5.5 days between the ascending and descending passes. The dates listed in the lower plot are at the middle of the 5.5-day period between the ascending and descending passes.**

