# Peer review of "A revised calibration of the interferometric mode of the CryoSat-2 radar altimeter improves ice height and height change measurements in west Greenland."

_The Cryosphere, 2016_

## Referee Comment (RC1) · Anonymous Referee #1 · 11 Jan 2017

This paper gives details of improved processing techniques for CryoSat-2 data, using swath-processed data to illuminate some apparent errors in the released data, and demonstrating the use of swath-mode data to monitor supraglacial lake heights in Greenland. The paper will be primarily of interest to radar altimetry specialists, but provides useful information about how to handle the CryoSat dataset and should improve the glaciological community' ability to make sense out of these data. The authors provide good evidence for their conclusions about the dataset, and show interesting examples for how the data can use used to measure lake-level changes.

The writing is generally clear, although I would recommend a careful reading and the addition of commas to some of the longer sentences. The figures, for the most part, make good use of space, and provide a good illustration of the physical principals at work. I make some editorial comments below, which should be relatively easy to correct.

Throughout: "skidoo" should be "Ski-Doo" or "snowmobile"

Section 3: Give a citation for the discussion of look angle and roll angle, either to the CS2 documentation or to a paper.

6:27 Define the look angle (relative to nadir?)

7.25: It would be helpful to say "in this section" rather than "here"

8.10: should be "change was small"

8.15-30: Experiments should be described in the past tense, results can be described either in past or present tense.

8.21: add comma after "further"

Section 3.3: This section needs some introductory material about the difference between the retrackers available from ESA and the retracker used here. Without this discussion, readers not familiar with the details of CS2 tracking are likely to be confused by the comment at 10:3-4 about retrackers that use all of the waveform.

10:5-10: Could phase ambiguity play a role in the location differences between the two products? If not, should explain why.

10: 11-12: Why not edit the L2 data based on coherence and return power? Both are available through comparison with the L1B data.

10:24: Add a hyphen: "roll-angle"

10:24: Should point out that there is a minimum error at about 0.007 degrees

10:24-26: These sentences are hard to read: I would break this into shorter sentences, to say that the POCA points are often local high points on the surface; if the beam is shifted a small distance in the across-track direction, the height error is small, while if the beam is shifted a larger distance, the assumed location is no longer on the flat part of the local high, and the error increases.

10:20-30: Are these ascending or descending passes? Both? Do the results in Fig.10 depend on what type of pass is used?

11:4 "are giving" should be "give"

11:2-15: For each of these numbers, specify which of the heights is on top: e.g. "...the average ATM-POCA height difference is -0.16 m, with POCA higher than ATM.

12:3-12, and figure 12. I found 12 hard to interpret, in large part because the location map is too small to distinguish the colored lines. I recommend that the April 21 and July 14 plots be combined on the same axes, and the May 20 and August 12 plots, and that the size of the axes be increased to use the additional space.

12:6-7: Need to describe how elevations for the lakes were extracted from the swath data. Why was no retracker needed (see comment on 14:30).

12:15 should be "at higher elevations"

12:28: "could" should be "can"

13: 6: should be "relative strengths" 13:9-10- the dates can be given on the plot, and are not especially helpful in the text.

13: 11-15: Doesn't the decrease in elevation imply a slow drainage rate? See the literature on the "firn aquifer" for evidence that a lake can slowly drain through pathways that are invisible at the surface

13:16-25: This paragraph needs an introduction : explain that you are now talking about another set of lakes giving another demonstration of the mapping technique.

14:9: "would" should be "may"

14:13 : I don't understand this sentence: What does "realistic" mean in this context?

14:22- "can be" should be "was"

14:30: Why is a retracker not required? If you plotted power vs. range for the lake returns you would see the same kind of pattern you see for a POCA return, so why not pick the first slope for the lake return?

15:4—it's worth noting that sun-synchronous orbits are limited to +- 81 degress, so central Antarctica and the arctic sea ice would be missed.

Figure 7: To me, this looks like increasing longitude and increasing range are in opposite directions. It would be helpful to reverse the range panel so that the two are easier to compare. It would also be helpful to plot the ESA L2 range on the radargram.

Figure 9: It would be helpful to note the differences in x scale between the two rows of histograms.

———————————————————

---

## Referee Comment (RC2) · Anonymous Referee #2 · 1 Feb 2017

The manuscript by Gray et al., provides an analysis of CryoSat SARIn mode for the study of surface elevation over the Devon Ice cap and over a sector of the West Greenland Ice Sheet. The manuscript describes an improved calibration of the satellite's attitude and a fine tuning of the signal characteristic to obtain precise height measurements. There is then a detailed and informative discussion about the signal properties observed above supra-glacial lakes and of supra-glacial lakes elevation determination during a 6-year period.

The paper provides rigorous, thorough, novel and relevant observations of CryoSat's

performances over ice caps and ice sheets. My comments address the clarity of the manuscript that I think would benefit from improving on the following points:

The paper should make it clearer where the improvements in processing are (title) compared to previous work. As written only the improvement on calibration is explicit.

The paper should make it clearer when the roll-biased corrected heights have been used to generate bias and dispersion from airborne dataset.

Through the paper there is variability in the tuning of the various technical steps (e.g. processing's thresholds, filtering and binning) for different height products and area that often seems arbitrary. It would be good to have more details on the observations that led the authors to tune the processing the way they did so as to educate the reader.

The link between the calibration and the supra-glacial lake survey is not particularly explicit, the two parts are somewhat disjointed. What observation over the lakes led you to develop this improved calibration? I would like to see more on relating the impact of the better calibration with the gain in precision in the lake height measurement maybe via a dedicated section or by a better articulation of the two sections.

Here are detailed comments related to specific page and line numbers:

Title: CryoSat-2? ESA prefers this appellation.

Title: It is relatively clear that the paper provides an improved calibration however the improvement in processing upon previous work is not clear.

P1L15-16: How did you come up with these numbers? I could not find them in the manuscript.

P1L23: If I understood correctly this value of 0.5m is obtained from a consistency check between ascending and descending orbit path of CryoSat data, which can both be affected by a systematic bias, and not from validation with an auxiliary dataset. If so then I would rephrase this sentence and use a different word than accuracy.

P2L9: 'Point-Of-Closest-Approach'

P2L17: This is another nice piece of work on the subject: http://ieeexplore.ieee.org/document/7542661/

P2L20: and ice sheet margins

P2L21: not sure what 'borrows heavily' means?

P3L12: Hawley et al., GRL 2009 maybe also for a bit of history

P3L13: With respect to use of L1b product and generation of swath altimetry, it would be worth mentioning some recent work generating and applying swath altimetry to derive geophysical variables over ice sheets: Christie et al., GRL, 2016 (10.1002/2016GL068972), supra-glacial lakes: Ignéczi et al., GRL, 2016, (in the sup. mat. 10.1002/2016GL070338), Ice Caps: Foresta et al., GRL, 2016 (10.1002/2016GL071485)

P3L26: This sentence "Our method in working . . ." is awkward.

P4L12: What reference power is used to calculate the logarithmic values?

P4L10: Not sure to what this last sentence refers to.

P5Section2.2: What criteria do the authors used to identify swath returns from POCA returns? There seems to be a process by which these two records are identified within a waveform but the methodology/criteria to achieve this are not described.

P5L14: What distance is this? In the ground plane? Does the surface slope matters and how does that value varies with the slope? It is not clear how this values relates to the 4-bin filter described above.

P5L12-15: The binning step is unclear? As written it reads as if it is a consequence of the filtering, is it? Why do both? This section, and the need for binning/filtering, needs more justification.

P5L18: What are the gains of using this alternative setup for summer data? It would be good to describe further the motivation/justifications behind this customised processing.

P5L20: In Gray et al., 2013, the range of acceptable cross-track slopes for swath processing is between 0.5 and 2o. Are the authors revising this range? If so it would be interesting to have a paragraph or so discussing this.

P5Section2.3: What constrained was applied on the time difference between CS+ and validation data?

P8L16: fig. 5 is mentioned before fig. 4

P9L15: How was 0.0075o determined from the data in fig. 4/5? What is the uncertainty attached to this value?

P11L4: Is this after the bias correction is applied? How does this changes with a bias of 0.0075o applied?

P11L8: Why "appears"? Could this be checked?

P12 L14-18: A few references would be helpful in this paragraph.

P12L23: A sentence or 2 on the use of surface topography to map and model supra-glacial lakes is warranted here: e.g. GIMPDEM: Leeson et al., 2015 doi:10.1038/nclimate2463, and GIMPDEM and CryoSat-Swath DEM: Ignéczi et al., GRL, 2016 10.1002/2016GL070338

P13 L6 'for the six ascending passes'. Is that seven passes?

P13L15: The amplitude of the decrease is as large as the increase in both L1 and L2, albeit with a different timing - it seems therefore to be a significant signal. Could you expand on the relative differences and on the reason behind this signal? Especially since drainage as a cause is excluded.

P13L23: Sentence starting by "If we assume . . ." seems incomplete.

P13L31: Is it an advantage of swath or is it because of your choice to limit the swath data to small look angles (P5L20)?

P14L13: I would soften/rephrase this statement; Foresta et al., GRL, 2016 show that there are no significant differences between rates of height change from POCA and from Swath over Icelandic ice caps. Second the greater spatial coverage offered by swath can lead to measure of height change where POCA fails and so provide a 'better' solution (Smith et al, TC, 2016). Finally it depends on the surface slope (direction and magnitude). I would rephrase the paragraph using the studies of height change over Iceland and Thwaites to show existing evidences for the benefit of swath for height change measurements, and listing the various associated caveats.

P15Section6: Is a bullet-point conclusion appropriate for TC conclusions? I leave this to the editor to decide.

P15L16-17: Where did the paper demonstrate the relative accuracy?

Fig3. Transect in fig. 1 seems to be missing

Fig8. Specify time-period and/or CS2 passes used

Fig9. Same as for Figure 8

Fig10 '(A) 16 passes plotted. . .' There is only one plot on panel A. Delete '(top)' at the end of the first sentence since the panels are labelled A and B.

Fig15. Add latitude on y axis and possibly range window on x axes

Fig17. Units (m) are missing on both x and y axes

---

## Referee Comment (RC3) · Anonymous Referee #3 · 6 Feb 2017

The manuscript of Gray et al. discusses improvements in POCA and swath processing of Cryosat-2 data. Comparing the satellite data to elevations from airborne surveys, GPS transects and DEMs, they show that their approach of using the point of inflexion generally provides better results than the standard ESA POCA data for two regions. Further improvements are obtained by introducing an additional roll angle correction. Whereas the improved performance of slope-dependent and threshold retrackers has been discussed before in other studies (including some of the first author), the findings about the roll angle correction are important. The manuscript is well written and the presented analysis is meticulous and thorough and illustrated with appropriate illustrations, which make the paper accessible for non-radar specialist (although some background knowledge is still required). The application to lake height changes makes it interesting for the broader glaciological audience that The Cryosphere targets.

As one of the other reviewers also pointed out, section 4 feels a bit disjointed. It would be good to discuss if and how the improvements discussed in the other sections contributed to these results. Could these results be obtained with the standard data provided by ESA? The only weak part I could identify in the manuscript is section 4.1 on the effect of surface melt on SARIn waveforms. Although the reasoning is logical, this part is rather speculative ('We speculate. . .' appears 3 times) and doesn't add much to the paper. I would suggest to keep this part for a separate paper, backed up with in-situ/model data of the local snowpack characteristics (possibly at a different location where such data are available).

Most textual comments have already been raised by the other two reviewers. Below a list of additional suggestions:

P3L7: '[L2i contains] also the waveform': is this so? As far as I'm aware, the L2i data contains some waveform parameters (leading edge slope, max wavepower, etc.) but not the full waveform?

P5L14: 'binning and averaging the results in segments' : please clarify how you did the binning and averaging (e.g., the width of the segments).

P6L17: It's unclear to me how you estimated the error by looking at the cross-track slope. Or did you inspect the cross-track slope to remove bad measurements?

P7L5: I suggest to use a symbol for the phase. The ph might be interpreted as p times h.

P7EQ4: is this equation correct? Shouldn't it be -ph/KB + ph/KB(. . .)-. . .

P8L17: please provide separate numbers for the bias for ascending and descending passes

P8L30: what's the range of the sun elevation angle for the 2012 ascending passes?

P10L9-15: figure 8a (ESA L2) is based on 951 pairs, fig 8B (max slope retracker only 599). I assume that during the processing of the max slope retracker data, faulty data was already (partly) removed? For a fair comparison, numbers for the same pairs should be compared. NB: in figure 9A, there are suddenly 1608 pairs for the POCA data. Where does this difference come from?

P15L9: the choice for the 0.0075 deg bias is poorly motivated.

P10L28: two of the passes in fig 10B show a minimum std at $\sim$0.015 deg. Any idea why this is?

P13L25: SARIn data certainly has advantages, but it doesn't allow to estimate total lake volume as with Landsat/MODIS. Seems correct to point out this limitation.

P14L10: 'POCA better suited for temporal height changes': Forresta et al, GRL, 2106 recently estimated volume changes for Iceland using swath data and found that swath leads to more accurate results than POCA data. It would be good to briefly discuss how uncertainties in the swath data affect such estimates.

P15L6: a 73-day repeat would lead to a larger inter-groundtrack separation, i.e. a less dense coverage.

---

## Author Comment (AC1) · 7 Mar 2017

We would like to thank the three reviewers for the care and time they have taken in providing thoughtful and helpful reviews. Also, we would like to thank Tommaso Parrinello, ESA, and Michele Scagliola, Aresys, for providing new information on CryoSat relevant to one of the issues raised in our paper.
In the response below, the reviewer comments are in italics and our replies are in normal font.

**Anonymous Referee #1**

*This paper gives details of improved processing techniques for CryoSat-2 data, using swath-processed data to illuminate some apparent errors in the released data, and demonstrating the use of swath-mode data to monitor supraglacial lake heights in Greenland. The paper will be primarily of interest to radar altimetry specialists, but provides useful information about how to handle the CryoSat dataset and should improve the glaciological community' ability to make sense out of these data. The authors provide good evidence for their conclusions about the dataset, and show interesting examples for how the data can use used to measure lake-level changes.*

*The writing is generally clear, although I would recommend a careful reading and the addition of commas to some of the longer sentences.*

Some commas have been added.

*The figures, for the most part, make good use of space, and provide a good illustration of the physical principals at work. I make some editorial comments below, which should be relatively easy to correct.*

*Throughout: "skidoo" should be "Ski-Doo" or "snowmobile"*

Done; skidoo changed to snowmobile.

*Section 3: Give a citation for the discussion of look angle and roll angle, either to the CS2 documentation or to a paper.*

The key reference (Galin et al. 2013) has been included, and the comment added that the negative sign in equation 2 arises because the instrument is being used in the mode where transmission is from the left antenna, and both are used for reception. As pointed out in the Galin et al. paper, the relation between phase and the interferometric look angle should be $\beta = \pm\mathrm{asin}(\chi/kB)$, where the sign depends on which antenna is used for transmission. Since launch, the left antenna has been used for transmission, hence our use of the negative sign in equation 2.

*6:27 Define the look angle (relative to nadir?)*

Defined in lines 6:23 and 6:24, and explained further in the following paragraphs.

*7.25: It would be helpful to say "in this section" rather than "here"*

Done.

*8.10: should be "change was small"*

Changed.

*8.15-30: Experiments should be described in the past tense, results can be described either in past or present tense.*

Past tense is now used to describe both the experiment and the results.

*8.21: add comma after "further"*

Done.

*Section 3.3: This section needs some introductory material about the difference between the retrackers available from ESA and the retracker used here. Without this discussion, readers not familiar with the details of CS2 tracking are likely to be confused by the comment at 10:3-4 about retrackers that use all of the waveform.*

Additional text has been added at the appropriate place in section 3.3, and an additional reference has been included giving some details of the retrackers used in the ESA L2 products: This is; Buffard J.: CryoSat-2 Level 2 product evolutions and quality improvements in Baseline C. ESA report XCRY-GSEG-EOPG-TN-15-00004. Available from https://earth.esa.int/web/guest/document-library

*10:5-10: Could phase ambiguity play a role in the location differences between the two products? If not, should explain why.*

Yes, it could, but in this example the major problem is that the L2 retracker has been fooled by the peaks in the middle of the waveform. Figure 7A and B have been redone to provide more information. By working back from the L2 positions the phase can be calculated from the calculated look angle. Then the position in the waveform with this phase can be marked, this has been done in Fig 7A. This shows that the L2 retracker often picks a position in the middle of the waveform close to a local maximum. The appropriate text and figure caption have also been improved to make this clear.

*10: 11-12: Why not edit the L2 data based on coherence and return power? Both are available through comparison with the L1B data.*

Figure 8 has been redone using editing as suggested by the reviewer. Some of the L1b waveforms are rejected in our processing. The L2 results for exactly these waveforms are now also removed so that the comparison between the ATM-L1b and ATM-L2 height differences is as fair as possible. In Fig. 8A, the statistics do not include the 39 L2 height values with errors greater than 20 m, even so the L2 results are significantly worse than with processing from the L1b files.

*10:24: Add a hyphen: "roll-angle"*

When 'roll angle' appears as a descriptor, e.g. 'roll-angle bias', a hyphen has been added.

*10:24: Should point out that there is a minimum error at about 0.007 degrees*

The minimum error at ~ 0.0075° ±0.0025° has been added. Any time the roll-angle bias (or the equivalent phase bias) is referred to an estimate of the error in this value has been added.

*14:9: "would" should be "may"*

Changed.

*14:13 : I don't understand this sentence: What does "realistic" mean in this context?*

The offending sentence has been removed and this part of the 'Discussion' has been changed, also to satisfy comments from the other reviewers.

*14:22- "can be" should be "was"*

Corrected.

*14:30: Why is a retracker not required? If you plotted power vs. range for the lake returns you would see the same kind of pattern you see for a POCA return, so why not pick the first slope for the lake return?*

A retracker is required in the situation where a sequence of time samples include more and more of the surface. The 'retracker' is then needed to estimate when in time the surface return occurs. When looking at a lake return in the middle of a waveform one could also use a retracker, but there will always be multiple areas contributing to each point in the 'lake' area of the waveform. For this reason, we opted to use standard cross-track InSAR processing, and use the phase from each range pixel for mapping. Then the accuracy of an average height result can be estimated using the consistency of the height results from the strong returns in the waveform, which have been assumed to originate from a flat lake surface.

*15:4âˇAˇ Tit's worth noting that sun-synchronous orbits are limited to +- 81 degress, so central Antarctica and the arctic sea ice would be missed.*

The last two sentences in section 5, 'Discussion', have been removed. The ongoing discussion of a follow-on CryoSat mission is perhaps beyond the scope of this paper.

*Figure 7: To me, this looks like increasing longitude and increasing range are in opposite directions. It would be helpful to reverse the range panel so that the two are easier to compare. It would also be helpful to plot the ESA L2 range on the radargram.*

This figure has been improved, as described above, but in our opinion reversing the range window in Fig. 7A didn't make it easier to see the link between the POCA geocoding solution and the waveforms.

*Figure 9: It would be helpful to note the differences in x scale between the two rows of histograms.*

Figure 9 has been redone with the same x axis for all the histograms.

**Anonymous Referee #2**

*The manuscript by Gray et al., provides an analysis of CryoSat SARIn mode for the study of surface elevation over the Devon Ice cap and over a sector of the West Greenland Ice Sheet. The manuscript describes an improved calibration of the satellite's attitude and a fine tuning of the signal characteristic to obtain precise height measurements. There is then a detailed and informative discussion about the signal properties observed above supra-glacial lakes and of supra-glacial lakes elevation determination during a 6-year period.*

*The paper provides rigorous, thorough, novel and relevant observations of CryoSat's performances over ice caps and ice sheets. My comments address the clarity of the manuscript that I think would benefit from improving on the following points:*
*The paper should make it clearer where the improvements in processing are (title) compared to previous work. As written only the improvement on calibration is explicit. The paper should make it clearer when the roll-biased corrected heights have been used to generate bias and dispersion from airborne dataset.*

As the paper emphasis is on the improved SARIn mode calibration through a detailed comparison of the CryoSat results against surface height measurements, and less on processing methodology, we have changed the title to 'A revised calibration of the interferometric mode of the CryoSat-2 radar altimeter improves ice height and height change measurements in west Greenland'. But we have also tried to illustrate the effect of the improvements through the examples included, including the work on supraglacial heights.

*Through the paper there is variability in the tuning of the various technical steps (e.g. processing's thresholds, filtering and binning) for different height products and area that often seems arbitrary. It would be good to have more details on the observations that led the authors to tune the processing the way they did so as to educate the reader.*

We have attempted to do this in a number of places in the text. In particular, we found that the west Greenland site was more challenging than the west Devon site, due to the 'rougher' topography, and this lead to more stringent data editing. This is explained and documented.

*The link between the calibration and the supra-glacial lake survey is not particularly explicit, the two parts are somewhat disjointed. What observation over the lakes led you to develop this improved calibration? I would like to see more on relating the impact of the better calibration with the gain in precision in the lake height measurement maybe via a dedicated section or by a better articulation of the two sections.*

We acknowledge that this is true. Rather than answer the question 'what observation over the lakes led you to develop this improved calibration?', we have tried to show that the improved calibration, coupled with our methodology, allows more precise ice height estimates, including the heights of the supraglacial lakes.

*Here are detailed comments related to specific page and line numbers:*

*Title: CryoSat-2? ESA prefers this appellation.*

We have changed the title, although in recent ESA publications, and in the programme of the upcoming ESA CryoSat meeting, the '2' is often absent.

*Title: It is relatively clear that the paper provides an improved calibration however the improvement in processing upon previous work is not clear.*

The title has been changed.

*P1L15-16: How did you come up with these numbers? I could not find them in the manuscript.*

The numbers reflect many comparisons of surface and CryoSat heights, including other test areas. As the abstract should reflect only the work in the paper, we now use the standard deviations from Fig 9A and 9C to reflect the relative precision of POCA and swath results from the west Greenland site. The revised text is… "While individual swath processed heights are normally less precise than edited POCA heights, e.g. standard deviations of ~3 m and ~1.5 m respectively for the West Greenland site, the increased coverage possible with swath data complements the POCA data and provides useful information for both system calibration and improving digital elevation models (DEMs)."

*P1L23: If I understood correctly this value of 0.5m is obtained from a consistency check between ascending and descending orbit path of CryoSat data, which can both be affected by a systematic bias, and not from validation with an auxiliary dataset. If so then I would rephrase this sentence and use a different word than accuracy.*

The text has been changed to '… a height precision of ~0.5 m for…'. We have checked throughout the text to make sure that we haven't used 'accuracy' when 'precision' would be more appropriate.

*P2L9: 'Point-Of-Closest-Approach'*

Changed.

*P2L17: This is another nice piece of work on the subject: http://ieeexplore.ieee.org/document/7542661/*

This reference has been added.

*P2L20: and ice sheet margins*

This phrase has been added.

*P2L21: not sure what 'borrows heavily' means?*

The text has been simplified… "The new approach uses bursts of pulses in which… "

*P3L12: Hawley et al., GRL 2009 maybe also for a bit of history*

The reference, and an appropriate sentence, have been added.

*P3L13: With respect to use of L1b product and generation of swath altimetry, it would be worth mentioning some recent work generating and applying swath altimetry to derive geophysical variables over ice sheets: Christie et al., GRL, 2016 (10.1002/2016GL068972), supra-glacial lakes: Ignéczi et al., GRL, 2016, (in*

*the sup. mat. 10.1002/2016GL070338), Ice Caps: Foresta et al., GRL, 2016
(10.1002/2016GL071485)*

These references have been added.

*P3L26: This sentence "Our method in working : : :" is awkward.*

The sentence has been changed, and simplified.

*P4L12: What reference power is used to calculate the logarithmic values?*

The text has been changed to… The L1b files contain two echo scaling parameters for each waveform which allow a calibration of the waveform power to watts, the logarithmic (dB) values used in the results are then with respect to 1 watt.

*P4L10: Not sure to what this last sentence refers to.*

The sentence was included to provide the background information to show that 'cross-overs' are not appropriate for calibration of the SARIn mode results.
"The position of the POCA footprint derived from each waveform will be in the plane including the satellite position, and the lines defined by the cross-track and nadir directions. The POCA area will be centred on the closest point in the intersection of this plane with the terrain surface, so that when ascending and descending orbits cross the two POCA footprints will not be the same when there is a cross-track slope. Consequently, it is not appropriate to compare results from the interpolated orbital cross-over point."

*P5Section2.2: What criteria do the authors used to identify swath returns from POCA returns? There seems to be a process by which these two records are identified within a waveform but the methodology/criteria to achieve this are not described.*

Our earlier papers, and the work of others (e.g. the Smith et al., 2016 reference), provides the necessary background on this. In our algorithm if the average of the first 5 values in the waveform are above a certain level then no POCA value is calculated, but swath mode results may be. This has been added.

*P5L14: What distance is this? In the ground plane? Does the surface slope matters and how does that value varies with the slope? It is not clear how this values relates to the 4-bin filter described above.*

All the geocoded footprints from one waveform are mapped into the zero Doppler plane, so that the POCA footprint and any swath footprints from the same waveform will be mapped to a straight line on the ice surface, irrespective of the surface topography. This is a straightforward consequence of the Delay-Doppler processing on data from a yaw-steered satellite.

*P5L12-15: The binning step is unclear? As written it reads as if it is a consequence of the filtering, is it? Why do both? This section, and the need for binning/filtering, needs more justification.*

Low pass filtering of complex waveform data has been used and described in this and in previous work (the Gray, Smith and Nilsson references), the benefit is reduced phase noise and therefore

better POCA and swath mode footprint geocoding. A comment to this effect has been added to section 2.1.

After the waveform smoothing the results of swath processing can be spaced a few tens of meters apart in the cross-track direction depending on the cross-track slope, and, as explained, may be oversampled. Consequently, it makes sense to carry out further averaging for swath mode results, this has been done by binning the results after the geocoding stage. The nominal separation of the footprint centres is 100 m in the cross-track direction. The text has been improved to help make this clear.

*P5L18: What are the gains of using this alternative setup for summer data? It would be good to describe further the motivation/justifications behind this customised processing.*

The lake features are small and few in number, so it is simpler to avoid the swath mode binning stage and look at each set of strong returns separately.

*P5L20: In Gray et al., 2013, the range of acceptable cross-track slopes for swath processing is between 0.5 and 2o. Are the authors revising this range? If so it would be interesting to have a paragraph or so discussing this.*

The original range of acceptable slopes (average cross-track slopes of ~0.5° to ~2°, Gray et al., 2013) was based on modelling using antenna gains etc., and was supported by the results from the western slopes of Devon Ice Cap. Experience from other sites and further comparisons with surface elevations have shown that, not unexpectedly, the best results are obtained with a slightly smaller range. The text on P8L9 has been expanded as follows…

The western portion of Devon Ice Cap has suitable cross-track slopes for swath mode height estimation for both ascending and descending passes, and this area was used in the demonstration of swath mode processing (Gray et al. 2013). While the possible range of average slopes can be ~0.5° to ~2°, here we have restricted the use of results to E-W slopes of ~0.7° - 1.5° over a distance of >5 km as this range generally provides a better suppression of the ambiguous range contribution.

*P5Section2.3: What constrained was applied on the time difference between CS+ and validation data?*

The rational for the selection of the time period for the CryoSat results is provided… 'Virtually all the reference height data for both sites were obtained under cold conditions in April or early May and we assumed that any accumulation or change in the backscatter conditions between January and mid-May would lead to a relatively small change in the CryoSat height. This provided the rationale for comparing all the CryoSat results from the January to May passes with the April or May reference height data.'

However, we also redid Fig. 8 (to satisfy a comment from reviewer 3), and used a smaller time window for the CryoSat data. (passes from 16 Feb. to 23 Apr.). Although there were now fewer CryoSat height samples, the results were similar to the previous figure.

*P8L16: fig. 5 is mentioned before fig. 4*

The reference to Fig. 5C has been removed.

*P9L15: How was 0.0075o determined from the data in fig. 4/5? What is the uncertainty attached to this value?*

The value was estimated from the cross-over of the lines from the experiment in which roll-angle bias is varied between -.2° and +.2°. Clearly, the value of 0.0075° is an estimate, and it is also hard to quantify the uncertainty as it will involve which star-tracker was used, and the position in the orbit. We have used an uncertainty of ±0.0025° in this value and explained that this just an estimate based on a relatively small sample.

*P11L4: Is this after the bias correction is applied? How does this changes with a bias of 0.0075o applied?*

Yes, the roll-angle bias has been applied, and the text on P10L16-19 states this. Also, the caption for Fig. 9 make it clear what has been done.

*P11L8: Why "appears"? Could this be checked?*

We are convinced that our explanation of the height bias is correct, but have not used 'is' in place of 'appears to be' because it is hard to prove, and we haven't done so.

*P12 L14-18: A few references would be helpful in this paragraph.*

The references below have been added after the comment related to the use of DEMs.

*P12L23: A sentence or 2 on the use of surface topography to map and model supra-glacial lakes is warranted here: e.g. GIMPDEM: Leeson et al., 2015 doi:10.1038/nclimate2463, and GIMPDEM and CryoSat-Swath DEM: Ignéczi et al., GRL, 2016 10.1002/2016GL070338*

See above

*P13 L6 'for the six ascending passes'. Is that seven passes?*

Yes, six replaced by seven.

*P13L15: The amplitude of the decrease is as large as the increase in both L1 and L2, albeit with a different timing - it seems therefore to be a significant signal. Could you expand on the relative differences and on the reason behind this signal? Especially since drainage as a cause is excluded.*

While the melt water created through the warm temperatures in summer 2012 at the positions of L1 and L2 did not drain in the dramatic way that supraglacial lakes often drain in the ablation zone, the water may have percolated through the firn in a manner similar to the recharging of firn aquifers, as has been recently observed in SE Greenland. New text, and two key references to the existence and study of firn aquifers in Greenland have been added, but in the final analysis we simply can't be certain why the year-to-year height signals decrease as shown, or why the height change is different for L1 and L2.

*P13L23: Sentence starting by "If we assume : : :" seems incomplete.*

Fixed…  'If we assume a lake area of $2 \pm 0.5$ km$^2$ this implies a filling rate of ~0.2. $10^6$ – $2.10^6$ m$^3$ melt water added per day.'

*P13L31: Is it an advantage of swath or is it because of your choice to limit the swath*

*data to small look angles (P5L20)?*

The description on P5L20 refers to the methodology for supraglacial lake height estimation. In this case, good results will only be obtained when the lake is beneath, or very nearly beneath the satellite, i.e., the look angles are very small. In general, the most reliable swath mode results over glacial ice are also obtained when the look-angle is small, the coherence high, and the waveform power above a certain limit.

*P14L13: I would soften/rephrase this statement; Foresta et al., GRL, 2016 show that there are no significant differences between rates of height change from POCA and from Swath over Icelandic ice caps. Second the greater spatial coverage offered by swath can lead to measure of height change where POCA fails and so provide a 'better' solution (Smith et al, TC, 2016). Finally it depends on the surface slope (direction and magnitude). I would rephrase the paragraph using the studies of height change over Iceland and Thwaites to show existing evidences for the benefit of swath for height change measurements, and listing the various associated caveats.*

We have expanded the text at this point and added reference here to the Foresta et al. and Smith et al. papers. It should be noted that Foresta et al. used the L2 POCA results rather than values derived from the L1b files. Also, in the Smith et al. work 'meter scale biases', correlated over tens of kilometers but independent orbit-to-orbit, were partially corrected by combining with the POCA data.

*P15Section6: Is a bullet-point conclusion appropriate for TC conclusions? I leave this to the editor to decide.*

? We admit bullet form is unusual, but we think it is 'efficient'.

*P15L16-17: Where did the paper demonstrate the relative accuracy?*

As commented above, we have reviewed the complete manuscript to make sure that 'accuracy' or 'accurate' are not used where 'precision' and precise' would be more appropriate.

*Fig3. Transect in fig. 1 seems to be missing*

The transect has been added, and the captions modified.

*Fig8. Specify time-period and/or CS2 passes used*

The time period for the CS2 data has been added to the figure caption.

*Fig9. Same as for Figure 8*

As above.

*Fig10 '(A) 16 passes plotted: : :' There is only one plot on panel A. Delete '(top)' at the end of the first sentence since the panels are labelled A and B.*

Fixed.

*Fig15. Add latitude on y axis and possibly range window on x axes*

The caption has been changed so that the directions of the x and y axes (increasing range and increasing along-track position (N up) are clear.

*Fig17. Units (m) are missing on both x and y axes*

Units have been added.

**Anonymous Referee #3**

*The manuscript of Gray et al. discusses improvements in POCA and swath processing of Cryosat-2 data. Comparing the satellite data to elevations from airborne surveys, GPS transects and DEMs, they show that their approach of using the point of inflexion generally provides better results than the standard ESA POCA data for two regions. Further improvements are obtained by introducing an additional roll angle correction. Whereas the improved performance of slope-dependent and threshold retrackers has been discussed before in other studies (including some of the first author), the findings about the roll angle correction are important. The manuscript is well written and the presented analysis is meticulous and thorough and illustrated with appropriate illustrations, which make the paper accessible for non-radar specialist (although some background knowledge is still required). The application to lake height changes makes it interesting for the broader glaciological audience that The Cryosphere targets. As one of the other reviewers also pointed out, section 4 feels a bit disjointed. It would be good to discuss if and how the improvements discussed in the other sections contributed to these results.*

*Could these results be obtained with the standard data provided by ESA?*

It isn't clear which results are being referred here. The revised calibration could not have been done with the standard L2 data, nor the work on supraglacial lake height. So we think the answer is no!

*The only weak part I could identify in the manuscript is section 4.1 on the effect of surface melt on SARIn waveforms. Although the reasoning is logical, this part is rather speculative ('We speculate: : :' appears 3 times) and doesn't add much to the paper. I would suggest to keep this part for a separate paper, backed up with in-situ/model data of the local snowpack characteristics (possibly at a different location where such data are available).*

This section has not been removed, but has been improved. First, it is well known that the introduction of even a small amount of liquid water in snow dramatically alters the emissivity and backscatter (Ulaby et al., 1986). This is the basis of the well documented change in both microwave emissivity and backscatter with the onset of melt in snow over Greenland. For example, a significant drop in QuikSCAT 13.3 GHz backscatter was shown to be linked to melting from weather station data (Nghiem et al., 2001). The presence of water droplets in snow increases absorption, reduces the penetration depth, which in turn leads to an increase in brightness temperature and decrease in radar backscatter (Wang et al., 2016). With this background, and by including these references, we have provided a more convincing and less speculative link between the drop in CryoSat returns and the onset of melt in the snow. Further, we feel that the link between spikes in the summer waveforms, and specular reflection from a wet

surface, is sufficiently clear that it also warrants inclusion. In summary, we hope that this short section will inspire a graduate student somewhere to systematically explore the links between the largely under-appreciated CryoSat data, other active and passive airborne and satellite data, and surface measurements. For this reason, we would like to continue to include this short section in the paper.

Ulaby, F., Moore, R., and Fung, A.: Microwave Remote Sensing: Active and Passive, Vol. 2, Norwood, Massachusetts, Artech House, 816–920, 1986.

Nghiem, S. V., Steffen, K., Kwok, R., and Tsai, W.-Y.: Detection of snowmelt regions on the Greenland ice sheet using diurnal backscatter change, J. Glaciol., 47(159), 539-547, doi: https://doi.org/10.3189/172756501781831738, 2001.

Wang, L., Toose, P., Brown, R. and Derksen, C.: Frequency and distribution of winter melt events from passive microwave satellite data in the Pan-Arctic, 1998-2013, The Cryosphere, 10, 2589–2602, doi:10.5194/tc-10-2589-2016, 2016.

*Most textual comments have already been raised by the other two reviewers. Below a list of additional suggestions:*

*P3L7: '[L2i contains] also the waveform': is this so? As far as I'm aware, the L2i data contains some waveform parameters (leading edge slope, max wavepower, etc.) but not the full waveform?*

The sentence has been changed to… 'An additional L2i product is available which contains the same geocoded height solution as the L2 product, but also information on the waveform which can be used to help eliminate poor data and solutions.'

*P5L14: 'binning and averaging the results in segments' : please clarify how you did the binning and averaging (e.g., the width of the segments).*

The text re binning and averaging has been improved, as noted for reviewer 2.

*P6L17: It's unclear to me how you estimated the error by looking at the cross-track slope. Or did you inspect the cross-track slope to remove bad measurements?*

If the 'lake' had a slope in the range direction this would indicate an error. If the slope was significant the measurement was removed, but a slope of a few tens of centimeters over hundreds of meters was used to help estimate the overall height error.

*P7L5: I suggest to use a symbol for the phase. The ph might be interpreted as p times h.*

Done, the symbol $\chi$ is now used for phase.

*P7EQ4: is this equation correct? Shouldn't it be -ph/KB + ph/KB(: : :)-: : :*

A sign error was corrected in equation 4.

*P8L17: please provide separate numbers for the bias for ascending and descending Passes*

These numbers have been added… The histogram of the difference between the reference and CryoSat swath mode heights obtained with the pre-launch baseline estimate (1.1676 m, Bouzinac, 2012) showed a bimodal distribution and the average bias changed between ascending and descending passes, ~-0.5 and ~2.5 m respectively.

*P8L30: what's the range of the sun elevation angle for the 2012 ascending passes?*

Thanks to the work of Scagliola et al. (2017), tracking the sun elevation angle to infer whether solar heating was involved in the roll-angle problem is now unnecessary. The main problem in the roll-angle issue appears to be a problem in processing the star tracker data. Reference to plate bending as the cause of the roll-angle problem has been removed, and the references to the solar elevation angle. Appropriate text has been added and reference to the new work included.

*P10L9-15: figure 8a (ESA L2) is based on 951 pairs, fig 8B (max slope retracker only 599). I assume that during the processing of the max slope retracker data, faulty data was already (partly) removed? For a fair comparison, numbers for the same pairs should be compared. NB: in figure 9A, there are suddenly 1608 pairs for the POCA data. Where does this difference come from?*

Figure 8 has been redone such that results from exactly the same waveforms have been compared, as described in the comment to reviewer 1. The difference in the number of samples used between Fig 8A and 9A arises simply because data from more passes were used in Fig 9A. This makes no difference to the conclusions.

*P15L9: the choice for the 0.0075 deg bias is poorly motivated.*

The text around the additional roll bias has been improved, as described for reviewer 2.

*P10L28: two of the passes in fig 10B show a minimum std at _0.015 deg. Any idea why this is?*

We think the variation in the position of the STD minima arises because of the pass-to-pass error in the roll angle, due to the incorrect processing of the 'optical aberration' in the star-tracker algorithm. This arises due to the variation in the relative motion of the star-tracker with respect to the field of stars. The dominant time period for this problem is then the orbital period.

*P13L25: SARIn data certainly has advantages, but it doesn't allow to estimate total lake volume as with Landsat/MODIS. Seems correct to point out this limitation.*

This is clear, we present the CryoSat results not as a competitor to the use of optical satellites but as another tool which might complement the study of these lakes, and the variation in surface melt year-to-year around Greenland.

*P14L10: 'POCA better suited for temporal height changes': Forresta et al, GRL, 2106 recently estimated volume changes for Iceland using swath data and found that swath leads to more accurate results than POCA data. It would be good to briefly discuss how uncertainties in the swath data affect such estimates.*

This part of the Discussions has been expanded, as described in the response to reviewer 2.

*P15L6: a 73-day repeat would lead to a larger inter-groundtrack separation, i.e. a less dense coverage.*

The comments related to a follow-on mission have been removed as beyond the scope of this paper.